# Mitochondrial Flexibility of Breast Cancers: A Growth Advantage and a Therapeutic Opportunity

**DOI:** 10.3390/cells8050401

**Published:** 2019-04-30

**Authors:** Angelica Avagliano, Maria Rosaria Ruocco, Federica Aliotta, Immacolata Belviso, Antonello Accurso, Stefania Masone, Stefania Montagnani, Alessandro Arcucci

**Affiliations:** 1Department of Public Health, University of Naples Federico II, 80131 Naples, Italy; immacolata.belviso@unina.it (I.B.); montagna@unina.it (S.M.); 2Department of Molecular Medicine and Medical Biotechnology, University of Naples Federico II, 80131 Naples, Italy; mariarosaria.ruocco2@unina.it (M.R.R.); aliotta.fed@gmail.com (F.A.); 3Department of General, Oncological, Bariatric and Endocrine-Metabolic Surgery, University of Naples Federico II, 80131 Naples, Italy; antonello.accurso@unina.it; 4Department of Clinical Medicine and Surgery, University of Naples Federico II, 80131 Naples, Italy; stefania.masone@unina.it

**Keywords:** breast cancer, tumour microenvironment, mitochondrial reprogramming, oxidative phosphorylation, therapeutic strategies

## Abstract

Breast cancers are very heterogeneous tissues with several cell types and metabolic pathways together sustaining the initiation and progression of disease and contributing to evasion from cancer therapies. Furthermore, breast cancer cells have an impressive metabolic plasticity that is regulated by the heterogeneous tumour microenvironment through bidirectional interactions. The structure and accessibility of nutrients within this unstable microenvironment influence the metabolism of cancer cells that shift between glycolysis and mitochondrial oxidative phosphorylation (OXPHOS) to produce adenosine triphosphate (ATP). In this scenario, the mitochondrial energetic pathways of cancer cells can be reprogrammed to modulate breast cancer’s progression and aggressiveness. Moreover, mitochondrial alterations can lead to crosstalk between the mitochondria and the nucleus, and subsequently affect cancer tissue properties. This article reviewed the metabolic plasticity of breast cancer cells, focussing mainly on breast cancer mitochondrial metabolic reprogramming and the mitochondrial alterations influencing nuclear pathways. Finally, the therapeutic strategies targeting molecules and pathways regulating cancer mitochondrial alterations are highlighted.

## 1. Introduction

Breast cancers are the most common solid tumour in women; they represent an important cause of mortality and have an increasing incidence rate in Europe, Latin America, Asia and Africa [1]. Moreover, these solid tumour tissues have different immunohistochemical profiles, which are linked to different clinical behaviours, and are constituted by cancer cells and the tumour microenvironment getting in touch through bidirectional interactions [1]. In particular, immunohistochemical studies divided breast cancers into three major types with different related percentages and prognosis: estrogen (ER^+^) and progesterone (PR^+^) receptor—positive, human epidermal growth factor receptor 2 positive (HER2^+^), and triple negative breast cancers (TNBCs) (Figure 1).

Breast cancers both expressing ER and PR represent approximately 85% of all breast cancers and are further divided into two subtypes: luminal A, which includes ER^+^ and/or PR^+^ and HER2^-^ breast cancer, and is characterised by the low expression of Ki-67 proliferation marker, and luminal B, which includes ER^+^ and/or PR^+^, HER2^+^ (or HER2^-^) breast tumours, showing high Ki-67 expression and worse prognosis than Luminal A. Both HER2^+^ and TNBCs account for about 15% of breast cancers [1]. Receptor-positive breast cancers have the best prognosis, while TNBCs, which are the most heterogeneous type of breast cancer, have a high risk of recurrence and a shorter overall survival compared with the other two types [1].

Breast cancers are very heterogeneous tissues constituted by epithelial cancer cells and an abnormal tumour microenvironment such as blood and lymphatic tumour vessels, an extracellular matrix (ECM), and non-cancer stromal cells represented by endothelial cells, pericytes, immune cells, cancer-associated fibroblasts (CAFs), activated adipocytes, and mesenchymal stem cells (MSCs) [2]. Therefore, cancer cells and their microenvironment constitute a tissue that behaves similar to a complex and heterogeneous metabolic ecosystem, where cancer cells can reprogram their metabolism as a result of interaction with microenvironment components [3,4,5]. Besides this tissue and metabolic heterogeneity, nowadays it is well known that cancer cells belong to a very heterogeneous cell community that is well organised functionally and hierarchically; within this community, cells coexist and act together to sustain their survival in response to the various microenvironments [3]. For example, MCF-7 breast cancer cells belong to a cell population including “bulk” cancer cells (~85–95% of the population), progenitor cells (<5%), and cancer stem cells (CSCs) (<1%). In particular, progenitor cells and CSCs are very dangerous, as they behave as tumour-initiating cells (TICs) in vivo and can undergo metastasis. On the other hand, “bulk” cancer cells represent a cell population that is characterised by a low tumorigenic potential [3]. However, one of the most impressive hallmarks of breast cancer cells is their metabolic plasticity [6]. In particular, in breast cancer cells, glycolysis is the main reservoir of energy: this process is called the Warburg effect [7]. The Warburg effect, which characterises the metabolic phenotype of cancer cells, is associated with a shift from mitochondrial oxidative phosphorylation (OXPHOS) to glycolysis, even in the presence of high oxygen tension, and can provide the building blocks that are necessary for a rapid proliferation [6,8]. On the other hand, increasing experimental evidence highlighted the important role of OXPHOS in tumour growth and progression. In fact, if OXPHOS is suppressed, cancer cells show an impaired ability to grow in an anchorage-independent manner as a dramatic reduction in tumorigenic potential [9]. Additionally, tumour cells with defective OXPHOS become very sensitive to cytotoxic drugs [9].

Recent studies have demonstrated that cancer metabolism is not static, but rather can be changed by the cellular needs that are regulated by cellular interactions linked to tumour cell–microenvironment crosstalk [4]. Within this complicated cellular and metabolic network, mitochondrial metabolic pathways can be reprogrammed to modulate breast cancer growth, and consequent mitochondrial alterations generate signals that influence nuclear cancer pathways [10].

In this review, we discuss the role of metabolic reprogramming in breast cancer initiation and progression, focusing on the role of mitochondria in this process. Furthermore, we review recent studies about mitochondrial–nuclear crosstalk and therapeutic strategies targeting breast cancer signals involved in mitochondrial reprogramming.

## 2. Metabolic Flexibility of Breast Cancers

In physiological conditions, both glycolysis and OXPHOS cooperate to produce energy. In particular, 70% of the energy request is provided by OXPHOS, while glycolysis can satisfy energy demand by generating two adenosine triphosphate (ATP) molecules through the cytoplasmic glucose metabolism [11]. However, glycolysis produces pyruvate that, in the presence of oxygen, enters the mitochondria to be oxidised and produces 36 ATP molecules [11]. Other fuels, represented by fatty acids (FAs), ketone bodies, and amino acids can sustain OXPHOS, and under hypoxic conditions, the glycolysis levels are increased and balance the reduction of OXPHOS activity [12]. Even if glycolysis is less efficient than OXPHOS in terms of ATP production, cancer cells through glycolysis can generate energy, by producing ATP more quickly and can also synthesise many cellular components required to grow [13].

Warburg supposed that the enhanced glycolysis in cancer cells was associated with the damage of mitochondrial respiration [10]. In fact, he attributed this metabolic switch to mitochondrial “respiration injury”, and considered this to be the most fundamental metabolic alteration leading to malignant transformation and to “the origin of cancer cells” [14]. It is known that tumour cells with a preponderant glycolytic metabolism are more malignant [15], and OXPHOS significantly influences Bax and Bak activation and cell death in cancer cells [16]. Hence, the metabolic switch from OXPHOS towards glycolysis is a means by which cancer cells protect themselves from programmed cell death and can sustain tumour progression [17]. However, recent studies of solid tumour metabolism changed the point of view regarding the role of OXPHOS in cancer metabolism, and showed that cancer cells have functional mitochondria with important functions in cancer initiation and progression [10,18]. Tan et al. showed by metastatic murine tumour models with mitochondrial DNA (mtDNA) deletion, that breast cancer cells depleted of mtDNA form tumours only after the acquisition of new mtDNA from tumour stroma and the recovery of mitochondrial respiration [19]. During cancer cell growth, the main function of OXPHOS is not represented by ATP production, but rather mostly by the synthesis of metabolites that are required for its proliferation. The mitochondrial electron transport chain (ETC) has an important role in cell proliferation, since it enables the synthesis of amino acid aspartate, which is a precursor in purine and pyrimidine synthesis [20,21,22]. In the mouse tumour model, pyrimidine biosynthesis, which is dependent on respiration-linked dihydroorotate dehydrogenase (DHODH), is essential for tumour growth, while the production of ATP by ATP synthase is dispensable for tumorigenesis [23].

Currently, cancer is considered not only a disorder of proliferation, but also a metabolic disease [24] in which cancer cells present a heterogeneous and dynamic metabolic profile: in fact, some cancer cells can use glycolysis as the main energetic pathway, while others can produce ATP mainly by OXPHOS [25,26,27]. In particular, tumour metabolic heterogeneity consists of inter-tumour (tumour by tumour) and intra-tumour (within a tumour) metabolic heterogeneity. In fact, there is a dramatic difference in the utilisation of glucose by cancer cells of different solid tumours and by different cancer cell populations of the same tumour (Figure 2) [4].

Concerning the inter-tumour metabolic heterogeneity, Lunetti et al. demonstrated that luminal-like and basal-like breast cancer cells display a distinct bioenergetic and metabolic phenotype, and consequently exhibit altered dependency on specific metabolic pathways (Figure 2A) [28]. Moreover, they also confirmed that distinct metabolic phenotypes are associated with different metastatic properties. The basal-like and highly metastatic MDA-MB-231 cell line exhibits a higher glycolytic flux and lower mitochondrial respiratory rate compared to the luminal-like MCF-7 cell line. MCF-7 cells are more dependent on mitochondrial respiration and have a low metastatic potential. The higher glycolytic rate and the reduction in OXPHOS metabolism that is observed in MDA-MB-231 cells establish their metastatic and invasive properties. The epithelial–mesenchymal transition (EMT), which is involved in the induction of cell motility and invasion by the disruption of cell–cell junctions, including the E-cadherin/β-catenin complex, is correlated to breast cancer metabolic state. In fact, the knockdown of E-cadherin and β-catenin induces a significant decrease in mitochondrial respiration and favours lactate production [28]. Regarding the intra-tumour heterogeneity, breast cancer cells display heterogeneity in metabolic choice associated with the simultaneous presence of a heterogeneous cell population [29]. In particular, the organisation of breast cancers follows a clear hierarchy, and the breast cancer stem cells (BCSCs) are at the top of this hierarchy. BCSCs are more dependent on OXPHOS, produce less lactate, have a higher number of mitochondria or more functional mitochondria, and have higher ATP content compared to their differentiated progeny. Surprisingly, BCSCs use more glucose than their differentiated progeny [29]. The role of mitochondrial metabolism in breast cancer cell migration and invasion is controversial: some studies linked glycolysis to EMT and consequently to the increase of breast cancer cell metastatic and invasive capability [28], while others reported a direct correlation between mitochondrial respiration and cancer cell metastasis. In fact, some evidence revealed that non-metastatic breast cancer proliferating cells satisfy their metabolic needs primarily through glycolysis [30], while in invasive metastatic breast cancer cells, mitochondrial respiration is induced to increase ATP levels through a mechanism associated with the overexpression of peroxisome proliferator-activated receptor gamma coactivator-1 alpha (PGC-1α), which mediates mitochondrial biogenesis [31]. Clinical data analysis of human invasive breast cancers showed a strong correlation between the expression of PGC-1α in invasive cancer cells and their ability to form distant metastasis [31]. Other findings suggested that metastatic breast cancer cells can use both glycolysis and OXPHOS metabolism, but, after metastatic colonisation, they choose a preferential metabolic program associated with site-specific metastasis [32]. The increase of PGC-1α expression is observed in breast cancer cells that metastasize preferentially to the lung and bone (Figure 2B) [33]. Conversely, liver metastatic breast cancer cells display an increase in glycolytic pathways and a decrease in glutamine metabolism and OXPHOS [32]. Breast cancer cells that successfully metastasize to the liver express high levels of pyruvate dehydrogenase kinase 1 (PDK1) (Figure 2B), which is activated by hypoxia-inducible factor (HIF)-1α and inhibits the function of pyruvate dehydrogenase (PDH), which is a key rate-limiting enzyme that is involved in pyruvate conversion to acetyl-CoA and entry into the tricarboxylic acid (TCA) cycle [32]. PDK1 silencing severely impairs the ability of breast cancer cells to metastasize to the liver, while their ability to form lung and bone metastasis is not affected [32]. Taken together, all these findings suggested that the metabolic heterogeneity and flexibility of breast cancer cells may dictate successful colonisation and metastatic cancer growth at distinct and specific sites [32].

Proteins that regulate glycolysis and mitochondrial metabolism are expressed in TNBCs, and the choice between glycolysis and mitochondrial respiration depends on the type of TNBC [34]. TNBC cells rely mostly on glycolysis to maintain cellular homeostasis. Additionally, TNBC cells can favour mitochondrial respiration in order to promote migration and distant metastasis [35]. Hence, several processes involved in the metastasis formation, such as invasion and migration, can favour mitochondrial biogenesis and OXPHOS. Furthermore, the metabolic diversity of cancer cells is also associated with the dramatic heterogeneity of the tumour microenvironment [2,4,36,37,38,39]. The variability of solid tumours such as breast cancer depends on the interactions between cancer and stromal cells as well as on blood supply. Cancer cells that are located close to the vasculature benefit from the availability of nutrients and oxygen, can produce ATP via aerobic OXPHOS, and trigger the anabolic pathway. On the other hand, when tumour mass increases and desmoplasia wrecks the organ’s normal architecture, new unsuitable vessels are built, and cancer cells get in touch with a microenvironment that is deprived of oxygen and nutrients, and has increased acidity [2,4,36,37,38,39]. In this situation, cancer cells adapt to the harsh microenvironment by the upregulation of glycolytic enzymes leading to the glycolysis pathway [40]. In other words, there is a nutrient and oxygen gradient in the breast tumour microenvironment that affects the metabolic choices of cancer cells [4,41]. Therefore, taken together, all this evidence indicates that in a solid tumour, some cells can be glycolytic, while others display an energetic metabolism that is essentially supported by OXPHOS [26,27]. The effect of this metabolic heterogeneity is the metabolic connections of different cells leading to cell proliferation and tumour growth [4]. In this context, mitochondria are important regulators of the metabolic plasticity of cancer cells, and mitochondrial metabolic reprogramming has a critical role in cancer growth, stemness, and therapy resistance [10]. In particular, the reprogramming of mitochondria affects both the phenotype of cancer cells and resistance to chemotherapy [42].

At the present time, it is known that p53 acts not only as a tumour suppressor, but also regulates breast metabolic reprogramming [43]. Particularly, increasing experimental evidence has suggested that p53 regulates the equilibrium between glycolysis and mitochondrial respiration [44]. In fact, in breast cancer, p53 modulates the proteins that are involved in the regulation of mitochondrial metabolism, and also induces FA oxidation and NADH and FADH2 production, which triggers OXPHOS [43]. Moreover, p53 downregulates several factors of the glycolytic pathway such as the glucose-type transporters (GLUT) represented by GLUT1, GLUT3, and GLUT4, triggers the expression of the TP53-induced glycolysis regulator and glutaminase-2, which leads to an increase in the metabolite α-ketoglutarate; in turn, this induces mitochondrial respiration. Furthermore, p53 can trigger mitochondrial respiration by also upregulating the cytochrome C oxidase (COX) complex [43]. Therefore, the consequence of p53 activation is the repression of pyruvate overproduction, and the induction of mitochondrial respiration [43]. The dependence of breast cancer metabolism on hormone levels’ alterations [45] and the association of each breast cancer immunohistochemical type with different metabolic alterations are known [46]. In particular, ER-α binds directly with p53 and inhibits its function [47]. Moreover, p53 mutations induce alterations in both glycolysis and OXPHOS [45,46].

## 3. Mitochondrial Reprogramming in Breast Cancer Cells

Mitochondria are organelles whose morphology can be linked to energetic states and the vitality of cells [48]. The mitochondrial morphology is connected to the production of ATP through OXPHOS, the handling of programmed cell death, calcium homeostasis, and the production of reactive oxygen species (ROS) [49,50]. Furthermore, the regulation of mitochondrial morphology is associated with metabolic alterations (Figure 3) [49,51,52].

Mitochondrial fusion is connected to increased ATP production, whereas the repression of fusion leads to impaired OXPHOS, mtDNA depletion, and ROS production [49]. In addition, the equilibrium between fission and fusion is influenced by alterations of nutrient availability and metabolic requests: these processes cause mitochondria fragmentation or concatenation (Figure 3) [49]. In fact, some studies have shown that cells subjected to a nutrient-rich environment maintain their mitochondria in a fragmented state, while cells under starvation maintain their mitochondria in a concatenated elongated state [51,52]. Particularly, Xing et al. investigated the alterations in breast cancer cell mitochondrial morphology associated with nutrition deprivation [48]. This investigation demonstrated that MDA-MB-231 and MCF-7 cells, when cultured in low-glucose media, present mitochondrial elongation. This mitochondria elongation is regulated by protein kinase A (PKA) activation and the consequent phosphorylation and inactivation of dynamin-related protein 1 (DRP1), which is an important factor regulating mitochondrial fission/fusion machinery, namely the mitochondrial dynamics. Furthermore, this process is correlated with a switch from glycolysis towards OXPHOS and cancer cell survival under stress conditions [48]. DRP1 is upregulated in MDA-MB-231 breast cancer cells, compared to MCF-10A, non-tumorigenic human breast cells [53]. DRP1 upregulation is associated with reduced mitochondrial oxidative capacity and increased mitophagy with a consequent decrease of mitochondrial content or number. To balance mitochondrial number reduction, mitochondrial biogenesis is enhanced in breast cancer cells [53]. The paradoxical increase of both mitochondrial biogenesis and fission versus mitochondrial number decrease is due to an overactive process of mitochondrial turnover adopted by breast cancer cells. Moreover, the treatment of cancer cells with a DRP1 antagonist (Mdivi-1) eliminates mitophagy, metabolic reprogramming, and cancer cell viability [53]. These findings highlight the importance of both mitochondrial biogenesis and mitophagy for cancer treatment [53]. Endophilin A1, which is encoded by the SH3GL2 gene, is a vesicular endocytosis-associated protein that could be a breast cancer suppressor. In fact, SH3GL2 expression is frequently lost, and its downregulation is associated with breast cancer progression [54,55]. The overexpression of endophilin A1 in breast cancer cells is associated with its phosphorylation and translocation to mitochondria; it also enhances mitochondrial fusion and the expression of both mitochondrial fusion and biogenesis-associated proteins, such as Mitofusin 2 (MFN2) and PGC-1α. These processes lead to the activation of intrinsic apoptosis through the induction of superoxide (O_2_^−^) production and the release of cytochrome C (CYTC) from mitochondria to the cytoplasm, and are also connected with decreased lung and liver metastasis and primary tumour growth (Figure 3) [54,55]. However, the mitochondria of cancer cells can use many metabolic pathways represented by glucose, glutamine, and FA oxidation (Figure 4) [10]. 

For example, many studies demonstrated that FAs are the primary energy font for TBNCs [10]. Furthermore, the supply of FAs to total ATP turnover in breast cancer cells is greater than that of glucose or glutamine [56]. FAs are substrates for ATP and NADPH synthesis and modulate cellular signalling pathways [57]. Additionally, the intracellular FA pool is provided by de novo lipogenesis, the intracellular triacylglycerol (TAG) of lipid droplets, and its exogenous source is represented by the blood circulation or tissue microenvironment [58]. FAs can be stored in lipid droplets as TAG or phospholipids [59,60], or translocate to mitochondria, where they are broken down by FA β-oxidation [61]. Moreover, FAs’ synthesis could have an important role in the pathogenesis of breast cancer, because breast cancer cells present an increased expression of the multifunctional enzyme fatty-acid synthase (FASN) [58]. Obesity affects breast cancer properties, and is connected to cancer risk by acquired insulin resistance [58,62]. In fact, the increase of hepatic and muscular lipids increases the availability of intracellular diacylglycerol and ceramide [58]. This process damages insulin signalling and represses glucose uptake triggered by insulin, leading to an increase of insulin secretion by pancreatic β cells and the enhancement of insulin-like growth factor-1 (IGF-1) availability. Both insulin and IGF-1 induce proliferation and inhibit apoptosis [58]. Balaban et al. studied in vitro the interaction between breast cancer cells and lipid-loaded “obese” adipocytes, leading to the mitochondrial reprogramming of breast cancer cells [63]. In particular, they showed through co-culture and conditioned media approaches that breast cancer cells stimulate FA mobilisation from adipocyte TAG stores. Additionally, adipocyte-derived free FAs are transferred to breast cancer cells, leading to FA metabolism and mitochondrial oxidative activity via increased carnitine palmitoyltransferase 1 (CPT1A) and ETC complex protein levels. These processes are connected with the increase of migration and proliferation of cancer cells, depending on adipocytes FAs’ release being linked to the enzymatic activity of adipose triglyceride lipase (ATGL) and hormone-sensitive lipase (HSL) [63]. Moreover, the transfer of FAs to breast cancer cells is enhanced from “obese” adipocytes, and this process is associated with an increased induction of cancer cell proliferation and migration [63]. However, breast cancer cells such as MCF-7 and MDA-MB-231 differ in their intracellular usage of FAs and in response to palmitate-induced apoptosis. Particularly, in MCF-7 cells, FAs are used for mitochondrial oxidation, while in MDA-MB-231 cells, FAs are used for intracellular TAG synthesis. Therefore, these differences in intracellular FA usage support differences in the sensitivity to palmitate-induced lipotoxicity. In fact, MDA-MB-231 cells are highly sensitive, whereas MCF-7 cells are partially protected [63]. Obesity can lead to poor prognosis for breast cancer in postmenopausal women [64]. In particular, leptin is an endocrine factor that is associated with obesity and induces breast cancer cell growth and invasiveness [64]. MCF-7 cells treated with leptin show an increased oxygen consumption rate and an ATP production that is more dependent on OXPHOS compared to untreated MCF-7 cells that display a more glycolytic phenotype [64]. Furthermore, FA oxidation could be the alternative fuel that is triggered by leptin, because the levels of proteins involved in the metabolism of FAs such as FAT/CD36 and CPT1 are increased by leptin treatment. Leptin supports the use of glucose for biosynthesis and lipids for energy production. These metabolic reprogramming adaptations induced by leptin may provide benefits for MCF-7 growth and could contribute, together with other factors, to the worse prognosis of breast cancer in obese women [64]. Smith et al. demonstrated that mitochondrial thiol modification by iodobutyl triphenyl phosphonium (IBT) triggers a metabolic reprogramming in MDA-MB-231 breast cancer cells that is linked to a decrease of mitochondrial bioenergetics, the metabolites of the Krebs cycle, and adenine nucleotides. This study suggests that mitochondrial thiol modification modulates metabolism and inhibits anaplerosis by decreasing aconitase enzymatic activity and glutaminase C protein levels [65]. Moreover, da Cruz et al. reported a correlation between paternal low-protein (LP) intake and mammary cancer risk in female offspring. Using a murine experimental system, they showed that paternal LP intake leads to a reduced birthweight of daughters and the metabolic reprogramming of their mammary tissue and tumours. This phenotype is associated with an increased mammary cancer risk and amino acid metabolism alterations, particularly with an increase in the energy-generating substrate glutamate [66]. Glutamate generated from glutamine can enter the TCA cycle in order to produce energy and act as a carbon and nitrogen source for the biosynthesis of nucleotides, FAs, and amino acids. Moreover, both the mammary tissues and tumours of LP daughters, i.e., from fathers with a LP diet, display a rewired nutrient-sensing mechanism associated with the suppression of the AMP-activated protein kinase (AMPK) pathway with subsequent activation of mammalian target of rapamycin (mTOR) signalling [66]. mTOR is a serine/threonine protein kinase that promotes tumour progression by regulating cancer cell survival and growth [67]. Mammalian AMPK is a crucial energy sensor that is activated by low-energy cellular states linked to AMP/ATP and ADP/ATP-raising ratios [68]. It restores energy homeostasis under low-energy states by switching on alternative catabolic pathways that generate ATP. On the other hand, AMPK switches off anabolic pathways and other processes that consume ATP [68]. Moreover, mammary tumours display an altered expression of the microRNAs (miR-200c, miR-92a, and miR-451a) that are linked to the suppression of an energy-sensor pathway regulated by AMPK. It is noteworthy that the same miRNA alteration levels are detected in the breast tumours of women from populations with high rate of a low birthweight [66]. All together, these data demonstrated that LP daughters have an increased risk of developing a more aggressive mammary cancer, with tumours arising earlier and growing faster than in control offspring [66]. Stromal density increases the risk for the development of breast cancer [69], and enhanced breast density, as evaluated by mammography, confers a four to six-fold increased risk of breast cancer incidence across various subtypes [70,71]. This enhanced breast density is linked to an increased deposition of ECM proteins, such as collagen I [72]. A recent study demonstrated that the high density of the collagen microenvironment leads to the mitochondrial metabolic reprogramming of cancer mammary cells [6]. Mammary cell line 4T1, when grown in high-density collagen matrices, showed decreased oxygen consumption and glucose metabolism via the TCA cycle compared to cells cultured in low-density matrices. Under high-density conditions, mammary cancer cells present an upregulation of genes that are associated with oxidative glutamine metabolism. As a consequence, oxidative glutamine metabolism is enhanced, and glutamine is used as a fuel source to drive the TCA cycle. These metabolic changes are not caused by modifications in cellular proliferation, ROS levels, or environmental oxygen deprivation correlated with high-density collagen gel. All together, these data demonstrated that cancer cells exhibit a dynamic metabolic phenotype, which is also affected by matrix collagen density. This metabolic plasticity allows cancer cells to respond and survive to the changes occurring in the tumour microenvironment [6].

It is noteworthy that glutamine is an important metabolite for several cellular processes such as oxidative metabolism and ATP production [11], and that the request for glutamine during the rapid proliferation of cancer cells needs an additional extracellular source [73,74]. Glutamine is indispensable to cell proliferation and survival, and consequently, its loss induces cell death [75]. The knockdown of glutamine synthetase, which is an enzyme that is involved in glutamine formation, sensitises MCF-7, human epithelial cervical cancer cells (HeLa), and A549 cells to glutamine loss independently of the supplementation of nutrients, and inhibits cell proliferation [75]. Moreover, glutamine is regulated by both oncogenes and tumour suppressor gene metabolism, and this makes the glutamine pathway an excellent tool for therapeutic strategies targeting mitochondrial activity [76,77,78]. The glutamine pathway depends mainly on the mitochondrial enzyme glutaminase. Glutaminolysis is an anaplerotic metabolic pathway that supplies intermediates for the Krebs cycle by converting glutamine to α-ketoglutarate, which permits proliferating cells to utilise the TCA cycle for biosynthesis [79]. Particularly, in breast cancers, mitochondrial glutaminolysis permits an efficient production of ATP and produces a substrate for protein synthesis and biomass production [79]. In a recent study, Wang et al. revealed how cancer cells undergo a mitochondrial metabolic reprogramming, exploiting glutamine metabolism to sustain their growth and survival in the harsh conditions, such as hypoxia. They detected a significantly increased uptake of glutamine in MCF-7, HeLa, and 4T1 cells during hypoxic conditions. Hypoxia inhibits mitochondrial ETC and leads to the accumulation of reducing equivalents, such as NADH [75]. Interestingly, a previous study reported that mitochondrial dysfunction promotes cells to use glutamine carbon for acetyl-CoA production through the reductive pathway [80]. Metallo et al. demonstrated that several cancer cell lines, including breast cancer cells, use the reductive carboxylation of glutamine as the primary route for lipids generation. Moreover, a significant increase in reductive carboxylation activity is observed in cancer cells cultured under hypoxia [81]. According to this evidence, Li et al. hypothesised that under hypoxia, via the reductive pathway, proliferating cells increase the metabolic requirement for glutamine carbon to generate acetyl CoA for lipid synthesis, whereas glutamine nitrogen is used for nucleotide biosynthesis and is enriched in dihyroorotate and orotate, which are the precursors of uridine monophosphate (UMP) [75].

Mitochondrial biogenesis is dependent on both the mitochondrial and nuclear genome, and is influenced by pathways regulated by PGC-1, and activated by the mitogenic signal and metabolic stress [82,83,84,85]. Particularly, the expression of PGC-1α induces mitochondrial biogenesis and OXPHOS: these processes are linked to cancer cell migration and the outgrowth and survival of cancer cells presenting K-RAS oncogene ablation [31,86]. Moreover, PGC-1α confers metabolic flexibility to breast cancer cells, which can choose an alternate metabolic program when mitochondrial respiration is inhibited by metabolic drugs, such as metformin. High expression levels of PGC-1α in the presence of metformin induce an alternate source of ATP production through the stimulation of glycolysis and facilitate anabolic metabolism by diverting mitochondrial metabolites in anabolic reactions [33]. Moreover, in breast cancer cells, the expression of PGC-1β, which is a member of the PGC-1 family, is induced by the IGF-1 receptor signalling axis, leading to mitochondrial respiration and mitochondria biogenesis [87]. In particular, in MCF-7 cells, IGF/PI3K signalling enhances mitochondrial biogenesis by increasing the pool of new mitochondria and increasing OXPHOS to produce the ATP that is needed for proliferation [42]. On the other hand, this activation signalling simultaneously degrades old, damaged mitochondria through the induction of BNIP3 expression, which is a mitophagy regulator. Therefore, the IGF-1 signalling could be essential for supporting cancer cell viability through the stimulation of PGC-1β-mediated mitochondrial biogenesis, and BNIP3-mediated mitochondrial turnover [42]. However, the relevance of mitochondria for solid tumours originates also from an investigation of mtDNA depletion [10]. In particular, Hayashi et al. and Cavalli et al. showed that HeLa cell tumorigenicity, which is lost after the depletion of mtDNAs, is recovered after the reintroduction of mtDNAs, and that breast cancer cells that are depleted of mtDNA present a reduced tumorigenic phenotype [88,89]. Therefore, these studies regarding the depletion of mtDNA led to the development of a transmitochondrial cybrid experimental model, which is useful for investigating the interactions between different mitochondria and a unique nuclear environment [10]. In particular, Kaipparettu et al. used the transmitochondrial cybrids and multiple OMIC approaches to understand mitochondrial reprogramming and mitochondria-regulated cancer pathways in TNBCs. This study showed the importance of mitochondrial reprogramming, especially in relation to mitochondrial FA oxidation, in c-Src protooncogene activation and metastasis [90].

## 4. Mitochondrial Retrograde Regulation

Mitochondria, the powerhouses and metabolic factories of mammalian cells, are fully integrated into the whole cell regulation and coordination mechanisms both to ensure cellular homeostasis and decide cellular fate [91]. It is known that mitochondria are not only bioenergetic and biosynthetic organelles, but also regulate signalling pathways that are associated with ROS, acetyl CoA, and α-ketoglutarate generation [92,93]. ROS levels, and consequent redox signalling, modulate both cell proliferation and inflammation processes [92,94]. On the other hand, acetyl-CoA and α-ketoglutarate levels are connected to the regulation of DNA transcription [95,96,97]. Therefore, the mitochondria continuously communicate with the nucleus, generating an extensive bidirectional crosstalk. Consequently, as long as nuclear genes exert direct control over mitochondrial gene expression and posttranslational modifications through anterograde signalling, the mitochondrial genome enables individual mitochondria to respond to changes in membrane potential [91], in the levels of NAD^+^/NADH, ROS, cytosolic Ca^2+^, and the ATP/AMP ratio [98], by expressing their own genes and allowing the regulation of nuclear gene expression [91]. The mitochondrial to nucleus connection was first described in yeast by Butow et al. in 1987 [99] and later in other organisms [100]. Parikh et al. demonstrated that yeast cells are able to respond to the quality and quantity of mtDNA and modulate the levels of nuclear-encoded RNAs as a means of intergenomic regulation [99]. The general process of this intergenomic crosstalk is conserved from simple eukaryotes, such as yeast, to humans, even if the specific molecular mechanisms, by which signal transduction occurs, vary across the species [101]. This pathway of communication that relays information regarding the state of mitochondria to the nucleus has been dubbed mitochondrial retrograde regulation (MRR) (Figure 5) [98].

MRR involves multiple factors, such as metabolic cues or more direct routes such as mitochondria-related changes in intracellular Ca^2+^ homeostasis that sense and transmit mitochondrial signals, which culminate in wide-ranging changes in nuclear gene expression [100]. It is known that metabolic enzymes can translocate to the nucleus and connect metabolism with DNA and histones methylation or histones acetylation, thus regulating DNA transcription, replication, and repair [102]. The mitochondrial pyruvate dehydrogenase complex (PDC), mediating pyruvate oxidation after glycolysis and fuelling the Krebs cycle, can contribute to mitochondrial dysfunction [103]. In particular, PDC can translocate to the nucleus of cancer cells where it produces acetyl CoA, which is utilised for the acetylation of histones that are essential for G1–S phase progression and the expression of S phase markers. It is noteworthy that the ETC Complex I inhibitor rotenone enhances the nuclear translocation of PDC, suggesting that mitochondrial dysfunction can induce the nuclear translocation of PDC. Therefore, PDC represents a link between metabolism and epigenetic or cell-cycle regulation [104]. DNA and histone methylation/demethylation takes place in the nucleus through the methyl group donor S-adenosylmethionine (SAM), methyltransferases, and demethylases [105]. Mitochondrial enzymes may also modulate nuclear methylation. The product of the wild-type enzyme isocitrate dehydrogenase (IDH), α-ketoglutarate, is a cofactor for many histone and DNA demethylases. Conversely, the significant accumulation of the oncometabolite 2-hydroxyglutarate produced by mutated mitochondrial IDH is a competitive inhibitor of these demethylases. Hence, the levels between normal metabolites and oncometabolites regulate the methylation in several cancer types. It is important to note that the oncometabolites can regulate the transcriptional activity of HIF or inhibit enzymes affecting DNA repair in cancer [105]. Either way, these changes lead to a reconfiguration of metabolism [101] or an attempt of cellular adjustments to an altered mitochondrial state [100]. MRR occurs under both physiological and pathological conditions [98,100], influencing a wide range of cellular activities, including nutrient sensing, growth control, aging, metabolic, and organelle homeostasis [101], cancer development, and cancer progression [100]. It is noteworthy that mitochondrial dysfunction is one of the most notable features of cancer cells, and can be caused not only by molecular alterations in nuclear genes, but also by mitochondrial genome defects [106], which have been described in almost all cancers [106,107]. Mitochondrial genetic status can impact nuclear genome stability in human cells, causing chromosomal instability (CIN), which has been reported as present in a variety of human tumours [107], including breast cancer [108,109]. Hence, mitochondria in cancer cells are altered both functionally and genetically [106]. In fact, several studies reported that the expression of mtDNA-encoded cytochrome c-oxidase subunit II, as well as mtDNA copy number, is decreased in breast cancer [110,111]. Furthermore, the depletion of mtDNA correlates not only with tumour progression, but also with the prognosis of breast cancer [112]. The direct role of MRR in cancer development and progression was demonstrated by mtDNA depletion studies and by generating transmitochondrial cybrid models [10]. Cavalli et al. demonstrated that breast cancer cells devoid of mtDNA exhibit a weak tumorigenic phenotype, showing impaired abilities to grow in an anchorage-independent manner and increased sensitivity to cytotoxic drugs [89]. Moreover, Delsite et al. identified several nuclear genes differentially expressed in breast cancer cell line (Rho^+^ MDA-MB-435 cells) and in its derivative depleted of mtDNA (Rho^0^ MDA-MB-435 cells) [113]. These nuclear genes are involved in cell signalling, cell architecture, energy metabolism, cell growth and differentiation, and apoptosis. Hence, this study provides the basis for future investigation of the pathways affected by the impairment of mitochondria in oncological conditions, which is unfortunately still not well known [113]. Apurinic/apyrimidinic endonuclease (APE1) is a multifunctional protein that is involved in the crosstalk between mitochondria and nucleus, and is often found downregulated in several tumours such as breast, ovary, prostate, melanoma, lymphoma, lung, and colon cancer. APE1 is a DNA repair protein that functions as a transcriptional cofactor to stimulate the DNA binding activity of AP-1 (Fos, Jun) proteins, as well as nuclear factor-*k*B (NF-*k*B), HIF-1α, HIF-like factor, paired-box 8 (Pax-8), and other proteins. APE1 also controls and interacts with p53. In vitro studies demonstrated that the depletion of mtDNA leads to APE1 downregulation and nuclear genome instability, which may cause increased tumorigenic changes [107]. Moreover, Singh et al. demonstrated that the tumorigenic phenotype can be a reversible event because APE1 expression and the anchorage-independent growth and invasion can be reversed to parental levels by transferring wild-type mitochondria to Rho^0^ cells. Taken together, these data suggest that APE1 plays a key role in mitochondria-mediated tumorigenesis at various organ sites, including the breast, ovary, prostate, melanoma, lymphoma, lung, and colon [107]. Kaipparrettu et al. clearly confirmed the partial reversibility of the cancer phenotype, demonstrating that the oncogenic properties of an aggressive cancer cell can be at least partially reversed by the crosstalk with non-cancerous mitochondria [114]. In fact, the mitochondria of non-tumorigenic cells are able to suppress several oncogenic pathways and make the cells less cancerous. Briefly, normal breast epithelial cells (MCF-10 cells) and breast cancer cells (MDA-MB-468 cells), which are both used as mitochondrial donors, were enucleated and fused with mtDNA-depleted metastatic osteosarcoma-derived 143B TK-cells, which are used as a nuclear donor to ensure a common nuclear background. Cybrid cells with non-cancerous mitochondria show significantly decreased cell proliferation in hypoxic conditions and a lower invasion index, form significantly less colonies compared to cybrids with cancer mitochondria, and downregulate many oncogenic pathways, such as the RAS, HER2, SRC, and p53 pathways. Additionally, if oncogenic pathways are downregulated, benign mitochondria are also able to increase the expression of tumour suppressor genes including RB transcriptional corepressor 1 (RB1), phosphatase and tensin homolog (PTEN), and von Hippel–Lindau (VHL) [114]. Moreover, in vivo tumour studies strongly confirmed the capability of non-cancerous mitochondria to mitigate the oncogenic potential by reducing the tumour weight and size [114]. Kulawiec et al. identified two major gene networks, fibronectin 1 (FN1) and p53, which were differentially regulated between parental and Rho^0^ breast epithelial cells. Bioinformatic analyses of the FN1 network identified laminin (LAMC2), integrin, and three of six members of peroxiredoxins (PRDXIII, PRDXIV, and PRDXV) whose expression were enhanced in Rho^0^ epithelial cells [110]. It is well known that ECM proteins play an active role in cancer development. In fact, an increase of laminin, integrin, and fibronectin is frequently observed in breast cancer [115,116]. In addition, the expression of PRDXs that are involved in oxidative stress response, especially PRDXIII, PRDXIV, and PRDXV, is increased in breast tumours [117]. In the p53 network, the expression of structural maintenance of chromosomes protein 4 (SMC4) and Werner syndrome ATP-dependent helicase (WRN) is downregulated and upregulated, respectively, suggesting that this network may affect chromosomal stability. In fact, DNA double-strand breaks are increased in Rho^0^ cells. Interestingly, tight junction proteins claudin-1 and claudin-7, which are often downregulated in primary breast tumours, are involved in the p53 network and significantly decrease in Rho^0^ cells. All these data suggest that mtDNA-encoded genes play a key role in the oncogenic transformation of breast epithelial cells, and that multiple pathways involved in MRR contribute to the transformation of breast epithelial cells [110].

Single-strand DNA-binding protein 1 (SSBP1), which is essential for mtDNA replication, is downregulated in several tumours, such as invasive ductal breast cancer, leading to mitochondrial dysfunction [118]. In fact, the loss of SSBP1 expression is associated with mtDNA copy number reduction, mitochondrial ROS overproduction, and cytosolic Ca^2+^ accumulation, which in turn activates calcineurin-dependent mitochondrial retrograde signalling. This retrograde pathway induces c-Rel/p50 nuclear localisation and consequently activates transforming growth factor β (TGFβ) transcription. TGFβ, through SMAD activation, promotes EMT, and increases breast cancer cell metastasis [118]. 

HIF-1α is a master regulator of the adaptive response to hypoxia that induces the transcription of genes such as vascular endothelial growth factor (VEGF), lysyl oxidase (LOX), GLUT1, and PDK1, which sustain vascularisation, metastasis, and tumour cell survival [119,120]. HIF-1α is upregulated in solid tumours even during normoxia, and can be associated with the mitochondria-to-nucleus connection [120]. In particular, the treatment of MDA-MB-231 cells with the membrane-permeable α-ketoglutarate analogue dimethyl-2-ketoglutarate (DKG) is associated with mitochondrial respiration uncoupled and the consequent accumulation of succinate and fumarate in breast cancer cells [120]. Therefore, the excess succinate/fumarate is transported out of the mitochondria into the cytoplasm, where it impairs the activity of HIF-α-prolyl hydroxylases (PHD), leading to the stabilisation and activation of HIF-1α, and creating a state known as pseudohypoxia [121,122], which increases the glycolysis and oxygen flux rates [120]. Furthermore, HIF-1α-induced transcriptional reprogramming in breast cancer cell triggers pluripotency markers, which contribute to tumour expansion in vivo [120].

The transmembrane protein 126A (TMEM126A) is a mitochondrial transmembrane protein that is anchored to the inner membrane close to the cristae [123], which is downregulated in highly metastatic breast cancer cells [124]. Low TMEM126A expression correlates with poor prognosis. TMEM126A downregulation triggers mitochondrial dysfunction, which is characterised by enhanced mitochondrial ROS production and the disruption of mitochondrial membrane potential (MMP) [124]. Mitochondrial ROS can be considered a secondary messenger that is able to induce MRR by activating downstream effectors [125]. Indo et al. presented evidence for signal transduction by mitochondrial ROS [126]. In particular, the loss of TMEM126A promotes cell adhesion and migration in breast cancer cells via ECM remodelling and EMT induced by mitochondrial ROS overproduction [124].

For the first time, Carden et al. reported the direct involvement of miRNA in breast cancer cells’ MRR. In particular, they demonstrated that miR-663 expression is regulated by mitochondrial ROS and mediates mitochondria to nucleus retrograde signalling. Dysfunctional mitochondria promote oxidative stress, increase the activity of methyltransferase, leading to a hypermethylated miR-663 promoter, and reduce miR-663 expression. miR-663 downregulation results in decreased expression of the nuclear-encoded OXPHOS subunits and OXPHOS enzymatic activities, which together promote breast tumorigenesis. The restoration of mtDNA and consequently of mitochondrial function in breast cancer cells reverses miR-663 to the parental level. miR-663 regulates the expression of OXPHOS assembly factors, directly interacts with the 3′-UTR of the Complex III assembly factor (UQCC2), promotes OXPHOS, and suppresses breast tumorigenesis [127].

## 5. Therapeutic Strategies Targeting Mitochondrial Alterations

Mitochondrial functions are modified in solid tumours because cancer cells need metabolic adaptation to the increased energy demand [128]. For this reason, several studies aim to improve the pharmacological treatment of breast cancer by developing mitochondria-targeted therapies that either re-establish the physiological function of mitochondria or trigger mitochondria-mediated cell death [129].

The metabolic alterations of solid tumours are usually connected to cancer cells’ dependence on exogenous amino acids [130]. In particular, exogenous arginine from diet is unnecessary for normal cells, while it is essential for the viability of cancer cells, whose dependence on exogenous arginine is termed arginine auxotrophy [130]. In fact, arginine is a semi-essential amino acid in humans, and plays an important role in cancer metabolism, since it is involved in the synthesis of nitric oxide, polyamines, nucleotides, proline, and glutamate [131]. Therefore, arginine is important for human cancer growth and in particular for solid tumours showing de novo chemoresistance and poor clinical outcome [131]. The arginine auxotrophy of many solid tumours is due to the downregulation of argininosuccinate synthetase (ASS1), which is a key enzyme in arginine biosynthesis [131]. Therefore, the decrease of enzyme expression in several solid tumours makes them sensitive to external arginine depletion [132]. Qiu et al. showed that arginine starvation, caused by exposure to pegylated arginine deiminase (ADI-PEG20), induces the autophagy-dependent death of ASS1-deficient TNBC cells, which are dependent on the uptake of extracellular arginine [132]. Moreover, arginine starvation triggers mitochondrial oxidative stress, which is linked to impaired mitochondrial bioenergetics and integrity, and kills breast cancer cells in vitro and in vivo only if they are autophagy-competent. Therefore, arginine restriction could represent a therapeutic strategy for patients affected by breast tumours showing low or absent ASS1 expression [132]. Additionally, Cheng et al. demonstrated that arginine starvation induces asparagine synthetase (ASNS) expression in breast cancer cells, thereby diverting cellular aspartate towards increased asparagine and disrupting the malate–aspartate shuttle, which is important for the electron transfer from the cytoplasm into the mitochondrial matrix. This inefficient malate–aspartate shuttle leads to mitochondrial dysfunction, which results in cell death induction. Beyond its role in the synthesis of proteins as well as purines and pyrimidines, aspartate is essential for proper mitochondrial respiration, since exogenous aspartate addition rescues cell viability during arginine starvation. Therefore, aspartate insufficiency, resulting from arginine starvation-induced ASNS upregulation, can be considered an important vulnerability in arginine-starved ASS1-low breast cancer cells [133]. Moreover, Birsoy et al. demonstrated that pyruvate supplementation allows cells with ETC dysfunction to proliferate by stimulating aspartate synthesis. In particular, pyruvate, by promoting the regeneration of NAD^+^ in the cytosol, activates malate dehydrogenase 1 (MDH1) to produce oxaloacetate and drive aspartate synthesis via the cytosolic aspartate aminotransferase GOT1. Consistent with these data, the antiproliferative effects of several ETC inhibitors are almost completely blocked by pyruvate addition [21]. Hence, taken together, all this evidence suggested that cells lacking ETC function depend on pyruvate/aspartate axis for their proliferation and survival.

Dichloroacetate (DCA) is a regulator of cellular metabolism that targets mitochondria and induces the activity of PDH, which shifts pyruvate metabolism from lactic acid formation to mitochondrial respiration [134]. Moreover, these processes stimulate mitochondrial dysfunction and trigger JNK signalling, which has been associated with apoptotic cell death in many experimental tumour systems [134,135,136]. Recent work highlighted the validity of photodynamic cancer therapy (PDT), which combines radiotherapy and immunogenic cell death (ICD) [137], even if PDT presents limited therapeutic value against deep-seated tumours [138]. PDT, using visible or near-infrared light with a photosensitiser, generates high levels of ROS, which efficiently affect both cancer cells and tumour microenvironment. In fact, PDT can exert direct cytotoxic effects on tumour cells, and reprogram the tumour microenvironment to exert a potent anti-tumour immune response [139]. Nawab Ali group showed that combined treatment of MCF-7 breast cancer cells with PDT and DCA inhibits cell growth, decreases mitochondrial membrane integrity, probably through ROS production, and enhances apoptosis [139]. Moreover, ICD could be considered as a potential cancer cell death mechanism induced by the anti-tumour combinatorial treatment with PDT and DCA. This study, showing the possibility of sensitising MCF-7 cells to both apoptosis and ICD, offers an additional therapeutic approach in ER^+^ breast cancer treatment. This novel therapeutic approach could overcome the non-immunogenic properties, metabolic transformation, and therapy resistance of cancer cells [139].

Matcha green tea (MGT) is a natural product that recent studies have described as a product with antiproliferative, anti-oxidant, anti-bacterial, and chemopreventive effects [140,141]. Bonucelli et al. showed that MGT could have therapeutic potentiality by modulating the metabolic reprogramming of MCF-7 cells. In particular, MGT inhibits the propagation of cancer stem cells’ (CSCs) population, suppresses both OXPHOS and glycolytic flux, and switches the metabolism of cancer cells towards a more quiescent state [140,141]. Moreover, proteomic analysis detected mitochondrial and glycolytic enzymes that are downregulated by MGT treatment. Thus, MGT could possess significant therapeutic potential by regulating the metabolic reprogramming of cancer cells [140,141].

It is known that aggressive tumours can use glutamine to increment ATP production and sustain lipid synthesis through the reductive TCA cycle. [142,143]. The ShcA adaptor protein regulates the cancer-signalling pathways downstream of receptor tyrosine kinases [144]. Im et al. demonstrated that breast tumours employ the ShcA pathway to increase their metabolic flexibility and enhance their glucose dependency. These processes are associated with the induction of glucose catabolism by both glycolysis and OXPHOS, and make breast cancer cells remarkably subjected to glucose supply. In particular, ShcA upregulates the levels of PGC-1α, which regulates mitochondrial metabolism and allows breast cancer cells to efficiently metabolise both glucose and glutamine and satisfy cancer bioenergetic and biosynthetic requests [144]. The reduction of ShcA signalling decreases the metabolic rate of breast cancer cells and renders them more dependent on glutamine metabolism, and thus more vulnerable to the mitochondrial metabolic inhibition exerted by biguanides. In fact, the anti-tumour properties of biguanides, such as metformin and phenformin, are consolidated in breast cancers with hindered ShcA signalling. Therefore, the ShcA/PGC-1α axis is an important regulator of breast cancer metabolic reprogramming, and impaired ShcA signalling sensitises breast cancers to biguanides that block Complex I of the ETC and make cells dependent on glycolysis [144,145,146,147,148]. Mitochondrial ETC, which is also called the respiratory chain, is localised in the inner membrane of mitochondria and is linked to ATP generation. The respiratory chain is formed by four associated membrane protein complexes called complexes I, II, III, and IV [149,150]. It is known that the aberrant activity of mitochondrial Complex I can modulate breast cancer progression, affecting tumour growth and metastasis [151]. Metformin is a drug that is extensively utilised in the treatment of type II diabetes mellitus, but recent studies have found a correlation between its pharmacological action and an anti-tumour effect [11,152]. Metformin acts as a mitochondrial toxin that inhibits OXPHOS by blocking Complex I of the ETC, which is required by breast cancer cells for energy production [11]. This inhibition indicates a reduction in ATP synthesis. Moreover, the reduction of metformin-induced mitochondrial activity leads breast cancer cells to take up more glucose, and this biological mechanism is associated with a therapeutic effect represented by a decrease of blood glucose levels [11]. Moreover, in vitro studies revealed that long-term metformin therapy induces acquired metformin resistance in cells belonging to the claudin-low subtype of TNBC. These resistant cultures become enriched in stem cell-like cancer (SCLC) cells, which are forced out of their normal mitochondrial metabolic state into a more glycolytic phenotype in response to pharmacological stress. SCLC cells exploit their metabolic plasticity to survive switching towards a more glycolytic state. However, this metabolic choice makes SCLC cells more similar to the more glycolytic non-stem-like cells [153], which have previously been shown to be susceptible to combined metformin and glycolytic inhibitors [154]. Metformin-adapted breast SCLC cells become sensitive to the inhibition of the C-terminal binding protein (CtBP) transcriptional repressors, which are potential therapeutic targets in highly glycolytic cancer cells [153].

It is known that in proliferating cells, the TCA cycle is an important source of biosynthetic precursors [155]. The metabolic intermediates that are necessary for anabolic processes have to be rapidly renewed, and both glucose and glutamine represent the most important carbon font of the anaplerotic reaction. In particular, the conversion of glutamine to glutamate, and then to the TCA cycle intermediate α-ketoglutarate, represents a mechanism for renewing the carbon sources that are lost from the cycle to anabolic pathways. The first reaction is catalysed by the mitochondrial enzyme glutaminase, while the second reaction is catalysed by glutamate dehydrogenase or by one of several transaminase enzymes [155]. Hence, glutamine utilisation is a common strategy used by cancer cells, including breast cancer cells, to produce enough ATP to support cell proliferation, and thus escape drug treatment [156]. For this reason, glutaminolysis and the glutamine transporters that are involved in glutamine metabolism have been proposed as potential drug targets that are able to restore sensitivity to the initial therapy in resistant cancer cells. SLC6A14, which is also known as ATB (0,+), is an amino acid transporter that is found upregulated in ER^+^ breast cancer, could be a novel and effective pharmacological target for breast cancer treatment. In fact, in vitro and in vivo studies demonstrated that the treatment of ER^+^ breast cancer cells with α-methyl-DL-tryptophan (α-MT), which is a selective blocker of SLC6A14, reduces tumour growth by starving the cells of glutamine, arginine, and essential amino acids, inhibiting mTOR and causing apoptosis and autophagy [157]. Moreover, CB-839, which is a selective inhibitor of glutaminase, was shown to have potent anti-tumour activity both in vitro and in vivo TNBC models. No antiproliferative activity was observed in an ER^+^ cell line treated with CB-839 [158]. Compound 968 is another glutaminase inhibitor with potent anti-tumour properties: in fact, compound 968 activates apoptosis, and decreases the invasiveness and resistance of MDA-MB-231 cells to the chemotherapeutic drug doxorubicin [159]. Furthermore, Lukey et al. reported that the transcription factor c-Jun, which is often found upregulated in a subset of human breast cancers, acts as a primary regulator of the mitochondrial glutaminase expression in human breast cancer cells. In particular, c-Jun activation leads to the induction of glutaminase gene expression and to the enhancement of glutaminase activity in TNBC cells. Moreover, c-Jun overexpression provokes dependence on the glutaminase reaction and confers sensitivity to the glutaminase inhibitor bis-2-(5-phenylacetamido-1,2,4-thiadiazol- 2-yl) ethyl sulfide (BPTES). Therefore, this study also confirmed that TBNC might be sensitive to therapeutic strategies targeting glutaminase [155].

Moreover, FA metabolism may be used as a potent anti-cancer target also. The inhibition of FASN in breast cancer cells causes the depletion of the end product long-chain FAs palmitate and the accumulation of the substrate malonyl CoA, which is a potent mediator of cytotoxicity, via the induction of apoptotic cell death. Interestingly, no comparable toxicity was detected in normal tissues, where malonyl-CoA accumulation may not be a significant problem, probably because the FASN pathway activity is normally low [160].

Tumour necrosis factor receptor-associated protein-1 (TRAP1) is an antiapoptotic protein belonging to the Heat Shock Protein 90 (HSP90) family that displays a predominant mitochondrial localisation, and is upregulated in many solid tumours where it has different roles [161,162]. However, the best well-known function of TRAP1 is protection against mitochondrial apoptosis by regulating the opening of the mitochondrial transition pore (MTP) through interaction with HSP90 and cyclophillin D [163]. This protein interacts directly with respiratory complexes, and contributes to the regulation of their stability and activity, but it is still unclear if such processes are linked to a reduced or increased respiratory capacity. In fact, TRAP1 can induce or inhibit OXPHOS, and the influence of this regulation on tumour growth is open to discussion. TRAP1 knockdown provokes a decrease of mitochondrial aerobic respiration, sensitises cells to lethal stimuli, and inhibits the in vivo tumour growth of both MDA-MB-231 and MCF-7 cells [164]. However, it is noteworthy that the inhibition of the tumorigenic capacity of breast cancer cells by TRAP1 is not associated with the repression of cell proliferation. Moreover, TRAP1 modulates the mitochondrial morphology. In fact, low levels of TRAP1 are linked to the rod-shaped mitochondrial phenotype in invasive and metastatic MDA-MB-231 cells, while high levels are associated with the tubular network-shaped mitochondrial phenotype in non-invasive MCF-7 cells. Interestingly, the expression of TRAP1 in human breast cancer samples is inversely associated with tumour grade. Furthermore, the overexpression of TRAP1 in MDA-MB-231 cells provokes mitochondrial fusion, induces mitochondrial tubular networks, and inhibits cell migration and invasion in vitro and in vivo. Therefore, TRAP1 modulates mitochondrial dynamics and function, and links these processes to the tumorigenesis of breast cancer. This evidence suggests a possible TRAP1 involvement in the development of therapeutic strategies targeting breast cancer [164]. Moreover, Amoroso et al. described the localisation of TRAP1 also in the outer side of the endoplasmic reticulum, where it protects from endoplasmic reticulum stress, checks the proteins destined for the mitochondria, and regulates intracellular proteins ubiquitination through interaction with the proteasome 26S subunit, ATPase 4 (TBP7) [165]. TRAP1 can be upregulated in human breast carcinomas, and this upregulation is correlated with endoplasmic reticulum stress [161]. Furthermore, TRAP1 function is important in favouring resistance to paclitaxel, which is a microtubule stabilising/endoplasmic reticulum stress inducer agent that is widely used in breast cancer therapy, and to genotoxic agents, such as anthracyclines [161,166]. Therefore, extramitochondrial TRAP1 could be targeted to revert chemotherapy resistance.

The angiogenesis of solid tumours is associated with interstitial hypoxia and sustains glycolysis [167,168]. Moreover, antiangiogenic agents may counteract the physiology of cancer blood vessels, normalise cancer tissue oxygenation, and ameliorate other features involved in tumour growth [167,169,170]. Several resistance mechanisms, mainly involving specific compensatory signalling loops, with potential clinical impact, have been described [171,172]. Resistance to antiangiogenic agents can be a major problem in cancer therapy, because these drugs, which have been approved by the Food and Drug Administration (FDA) for their use against solid tumours such as breast cancer, are widely used in oncology therapy [173]. Navarro et al. showed that in the murine metastatic breast cancer experimental model, multikinase inhibitor antiangiogenics (TKIs) induce hypoxia correction, switch towards mitochondrial metabolism, and make mitochondrial metabolism essential for tumour survival [173]. Therefore, pharmacological blockers of the nutritional stress response, such as two anti-mitochondrial respiration agents, phenformin and ME344, can inhibit tumour growth, leading to metabolic synthetic lethality in combination with TKIs [173].

Urra et al. identified a bromoalkyl ester of hydroquinone named FR58P1a as a mitochondrial metabolism-affecting compound that is able to decrease selectively the migratory capability of TNBC cells, without reducing breast cancer cell survival. TNBCs, which have a four-fold increased risk of developing distant metastasis, rely mostly on glycolysis to maintain cellular homeostasis, but use mitochondrial respiration to invade and metastasize distant sites. It is noteworthy that OXPHOS activity increases concomitantly with the metastatic potential of cancer cells. Therefore, mitochondrial respiration may represent a promising target for anti-metastatic therapy. Prolonged FR58P1a treatment reduces tumour migration by inducing a metabolic reprogramming in TNBC cells characterised by the downregulation of OXPHOS-related genes and a metabolic adaptation towards glycolysis [35].

Arif et al. identified a novel and potent therapeutic target for treating breast cancer and other solid tumours by modulating cancer metabolism. This molecular target is the mitochondria gatekeeper voltage-dependent anion channel 1 (VDAC1), which acts as a transporter for metabolites in and out of the mitochondria, including pyruvate, malate, succinate, nucleotides, NADH, cholesterol, and lipids. As a transporter of metabolites, VDAC1, which is upregulated in many tumours, controls cell energy and metabolic homeostasis [174,175], and regulates the metabolic phenotype of cancer cells [176]. In both in vitro and in mouse xenograft models of human glioblastoma lung cancer and TNBC, VDAC1 silencing leads to the inhibition of all the metabolism-related processes, including glycolysis, the TCA cycle, and OXPHOS. VDAC1 depletion is associated with a dramatic decrease in the expression levels of several glycolytic proteins, including GLUT1, glyceraldehyde dehydrogenase (GAPDH), hexokinase (HK1), and lactate dehydrogenase-A (LDH-A). Additionally, the expression levels of the Kreb’s cycle enzyme citrate synthase (CS) and Complex IVc are highly reduced in VDAC1-depleted cells. This results in a decrease in energy and metabolite generation, cell growth arrest, and the inhibition of tumour growth. These VDAC1 depletion-mediated effects involve alterations in the master transcription factors associated with metabolic control; in particular, they increase expression of p53 and decrease the expression of HIF-1α [176].

Tamoxifen (TAM) is a molecule belonging to anti-estrogens (AEs) that competes with endogenous estrogens for ERs’ interaction [1]. Koumenis et al. showed that TAM, beyond anti-estrogenic effects, can modulate metabolism through a non-ER, mitochondrial pathway in both ER^+^ MCF-7 cells and triple-negative MDA-MB-231 cells [177]. In particular, TAM interacts with the flavin mononucleotide site of Complex I, leading to OXPHOS inhibition and a significant decrease of cellular oxygen consumption [177,178]. The mitochondrial bioenergetic alterations, caused by TAM-induced Complex I inhibition, are also associated with an increase in the AMP/ATP ratio and activation of the AMPK signalling pathway both in vitro and in vivo, which induces glycolysis and modifies FA metabolism [177]. Since the inhibition of oxygen consumption by TAM often results in the upregulation of glycolysis, more successful strategies for cancer metabolic interventions may involve the inhibition of both OXPHOS and glycolysis simultaneously. In fact, the combined treatment of MCF-7 and MDA-MB-231 cells with TAM and glycolysis inhibitors, such as 2-deoxy-D-glucose and 3-bromopyruvate, significantly increases the cytotoxicity induced by TAM. Moreover, the sensitisation to TAM results in the MDA-MB-231 cells displaying a more evident higher basal level of glucose uptake and a lower basal level of OXPHOS [177]. However, even if TAM is currently used as a first-line treatment for hormone-sensitive breast cancers, it is inefficient in the HER2^high^ breast cancers, which are highly recalcitrant to therapy. For this reason, Rohlenova et al. designed and synthetised a new compound called MitoTam, which is a TAM derivative obtained through the attachment of a triphenylphosphonium (TPP^+^) group. The TPP^+^ group ensures the accumulation of MitoTam adjacent to Complex I, increasing its effects and specificity for mitochondria. In this regard, MitoTam was introduced into the family of mitocans, which are anticancer agents acting directly on mitochondria. MitoTam enhances ROS production and cell death, and kills TAM-resistant breast cancer cells. MitoTam-resistant breast cancer cells were not obtained also upon long-term culture (>6 months), suggesting that resistance to MitoTam may not develop. Moreover, MitoTam shows efficacy superior to that of TAM in the treatment of HER2^high^ breast cancers, both in vitro and in vivo. In particular, in a preclinical model, MitoTam was able to almost completely cure HER2^high^ breast cancers without deleterious side effects. MitoTam efficacy and toxicity are currently being evaluated in clinical trials [179].

Therefore, the next goal in breast cancer therapy is to understand if the combination of several antimetabolic therapeutic strategies could provide a potent addition to the anti-tumour armoury that is currently used to treat breast cancer. Metabolic compounds and therapeutic approaches targeting breast cancer metabolism, described in the text, are listed in the Table 1.

## 6. Conclusions

Although Warburg proposed that cancer cells have dysfunctional mitochondria and that mitochondrial damage might be the origin of cancer [14], several recent findings in cancer metabolism demonstrated that functional mitochondria are essential for cancer cell survival. In fact, most cancer cells not only have functioning mitochondria, but also use mitochondrial respiration to promote tumour growth and progression. In this regard, cancer cells do not inactivate mitochondrial energy metabolism, but rather alter the mitochondrial bioenergetic and biosynthetic state.

The collective evidence presented in this review highlighted the metabolic plasticity of breast cancer that can be considered as a heterogeneous metabolic disease, in which cancer cells may choose the best metabolic program to sustain tumour progression. In this scenario, mitochondria modulate the cancer bioenergetic plasticity that allows tumour cells to adapt and survive in the ever-changing and harsh tumour environment. In response to different microenvironments, hypoxia, or pharmacological stress, breast cancer cells can satisfy their bioenergetic and biosynthetic needs, choosing between glycolysis and OXPHOS and using many mitochondrial metabolic pathways, such as glucose, FAs, and glutamine oxidation [10]. Moreover, mitochondrial alterations, through MRR, actively control breast cancer progression by activating and sending oncogenic signals to the nucleus. This results in several changes in nuclear gene expression [91] and can lead to breast cancer initiation [110], progression, and dissemination towards distant sites [107,118,124,127]. Actually, therapeutic strategies targeting mitochondrial alterations in breast cancer cells are gaining increasing attention in oncology. Many agents targeting the specific enzymes involved in the metabolic pathways, such as glycolysis, glutaminolysis and FA synthesis, have been developed or proposed, and have been studied in clinical trials [156]. Mitochondria-targeted therapies may improve the pharmacological cancer therapy that is currently used in breast cancer patients, through either re-establishing the physiological anti-tumour function of normal mitochondria or triggering cell death induced by mitochondria [129]. However, long-term metabolic therapies targeting only one metabolic pathway could result in a rapid proliferation of resistant cancer cells and become ineffective against cancer. When a specific metabolic route is inhibited, the metabolic flexibility of breast cancer cells leads to the activation of alternative metabolic pathways that allow cancer cells to overcome drug sensitivity and survive. Hence, it could be necessary for the simultaneous use of drugs targeting different metabolic pathways to block the availability of all the bioenergetic sources in order to prevent successfully the metabolic adaptation and the consequent survival of breast cancer cells. On the other hand, we think that therapeutic strategies including compounds directed towards different metabolic pathways could be associated with side effects targeting the energy homeostasis of the organism. Another therapeutic option could be represented by targeting metabolic crosstalk pathways between cancer cells and CAFs that are characterised by a minor metabolic plasticity compared to cancer cells. Indeed, CAFs are metabolically enslaved by cancer cells to sustain tumour growth and nutrients demand [5]. Anyway, it is known that new anti-neoplastic therapeutic strategies should be based on the central role of mitochondria in cancer development [180,181]. Particularly, further investigations are required in order to better understand the molecular mechanisms involved in the mitochondrial reprogramming that influence the metabolic adaptation of cancer cells. Finally, in-depth comprehension regarding the molecular pathways involved in breast cancer metabolic plasticity would lead to developing a successful metabolic therapy that targets the different steps and progression of various breast cancer types.

## Figures and Tables

**Figure 1 cells-08-00401-f001:**
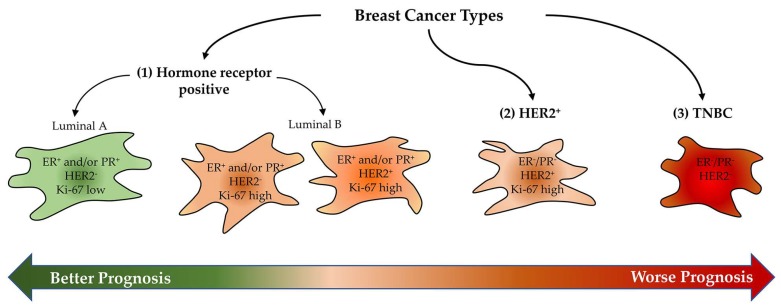
Classification of breast cancer into three major types based on their immunohistochemical properties and relative prognosis.

**Figure 2 cells-08-00401-f002:**
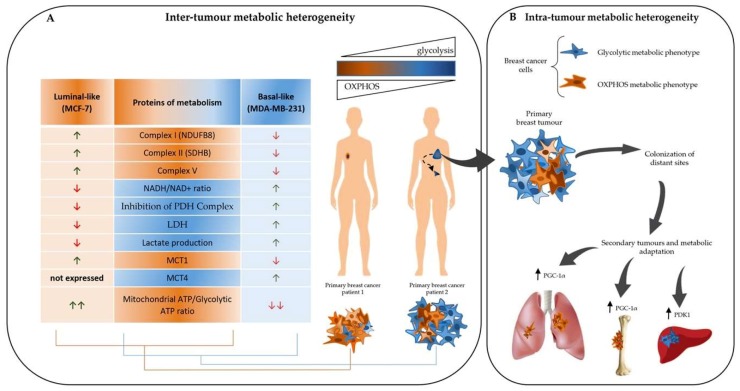
Breast cancer as a heterogeneous metabolic disease. (**A**) Inter-tumour metabolic heterogeneity refers to the ability of different breast cancer types to exhibit a distinct and preferential metabolic phenotype. MCF-7 cells, belonging to the luminal-like breast cancer subtype, are more dependent on mitochondrial respiration, and reduce lactate dehydrogenase (LDH) expression, and by consequence lactate production. MCF-7 cells increase monocarboxylate transporter 1 (MCT-1) protein levels to import the lactate produced by the tumour microenvironment into the cell. Conversely, the basal-like MDA-MB-231 cells exhibit a more glycolytic phenotype. The lower respiratory rate in MDA-MB-231 cells is associated with a strong decrease in complexes I, II, and V of the electron transport chain (ETC). As a result of dysfunctional Complex I, MDA-MB-231 cells exhibit higher levels of NADH, which in turn inhibits the pyruvate dehydrogenase (PDH) complex. Thus, pyruvate does not enter the tricarboxylic acid (TCA) cycle, but it is mainly converted to lactate by LDH, which is highly expressed in MDA-MB-231 cells. Lactate is efficiently extruded from MDA-MB-231 cells through the monocarboxylate transporter 4 (MCT-4), which is not expressed in MCF-7 cells. (**B**) Intra-tumour metabolic heterogeneity refers to the presence in the same tumour mass of a heterogeneous cell population displaying different metabolic phenotypes. Breast cancer cells are able to adapt their metabolism, according to several stresses and signals sent by the ever-changing and harsh microenvironment of both primary tumour and metastatic sites. In liver metastasis, breast cancer cells are more glycolytic and express high levels of pyruvate dehydrogenase kinase 1 (PDK1). In bone and lung metastasis, breast cancer cells preferentially use mitochondrial oxidative phosphorylation (OXPHOS) and upregulate peroxisome proliferator-activated receptor gamma coactivator-1 alpha (PGC-1α).

**Figure 3 cells-08-00401-f003:**
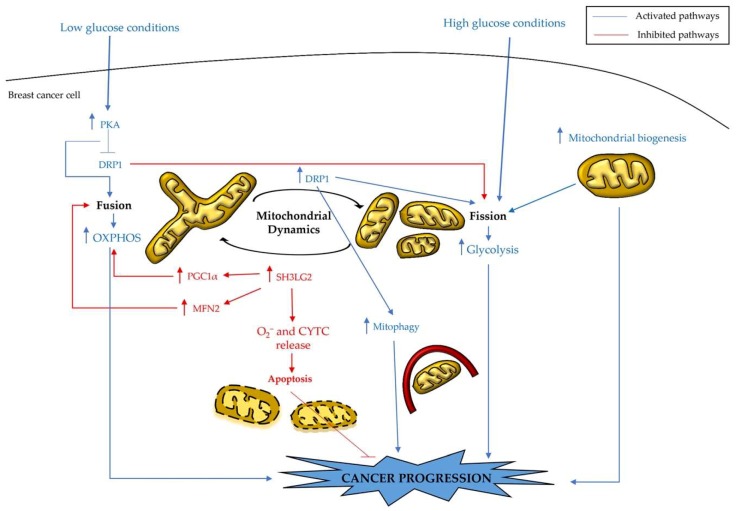
Mitochondrial morphology alterations and metabolic reprogramming. Mitochondrial morphology transitions (from elongated to fragmented, and vice versa) are linked to breast cancer metabolism. In fact, breast cancer cells alter mitochondrial dynamics to adjust their bioenergetics and biosynthetic needs in order to survive in harsh conditions and support tumour progression. In high-nutrient conditions, mitochondria are fragmented and dysfunctional, and energy demand is mainly supported by glycolysis. Dynamin-related protein 1 (DRP1) upregulation in breast cancer cells is associated with fission, glycolysis, and mitophagy. The reduction of mitochondrial number due to mitophagy is restored by an increase in mitochondrial biogenesis. The fission mechanism is also induced in breast cancer by the loss of the tumour suppressor gene, SH3LG2. When overexpressed, endophilin-A1 encoded by the SH3LG2 gene translocates to mitochondria, interacts with PGC-1α and Mitofusin 2 (MFN2), and triggers the mitochondrial fusion network and OXPHOS. Moreover, the mitochondrial translocation of SH3GL2 triggers the intrinsic apoptotic pathway through the induction of O_2_^−^ production and cytochrome C (CYTC) release. Conversely, in low nutrient conditions, breast cancer cells maintain their mitochondria in a hyperfused state by inhibiting the mitochondrial fission protein DRP1 and favouring energy production through OXPHOS. The blue arrows indicate activated pathways, whereas red arrows depict inhibited pathways in the breast cancer cell.

**Figure 4 cells-08-00401-f004:**
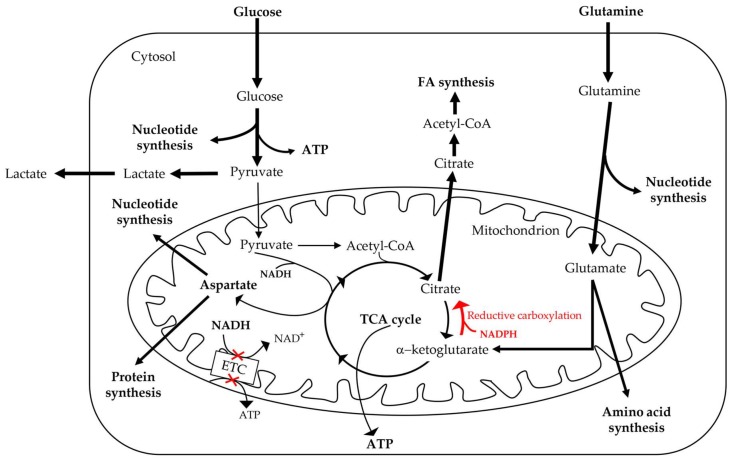
Functional alterations of mitochondria in breast cancer cells. Glucose carbon, which is mainly used for lactate production, is directed away from the TCA cycle and fatty acid (FA) synthesis. In this context, glutamine carbon feeds the TCA cycle and contributes to FA synthesis through the reductive pathway. The reductive carboxylation is supported by a disturbance in the redox ratio of mitochondria due to the aberrant function of the ETC, which decreases the NAD^+^/NADH ratio. This ratio is dissipated in part through the transfer of reducing equivalents from NADH to NADPH, which in turn may drive the NADPH-dependent reductive carboxylation of glutamine by isocitrate dehydrogenase 1 (IDH1) and IDH2. All these processes are further enhanced during hypoxia. Moreover, cells with ETC dysfunction use pyruvate, which is formed during glycolysis, to produce aspartate, which is important for cell proliferation and survival.

**Figure 5 cells-08-00401-f005:**
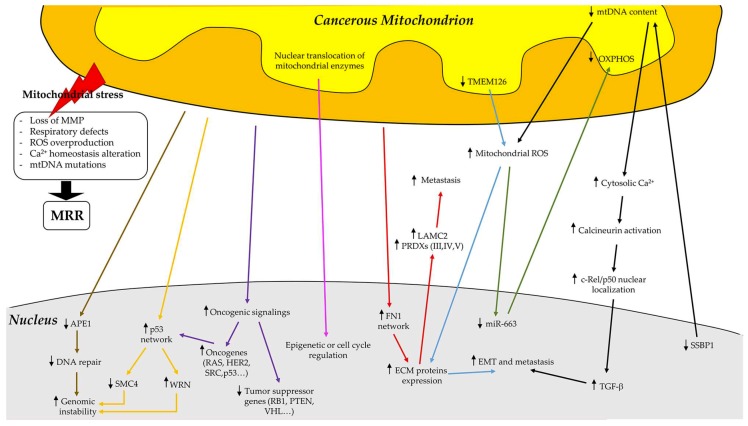
Mitochondrial retrograde regulation (MRR). Schematic representation of the main pathways used by cancerous mitochondria to communicate with the nucleus during breast cancer progression.

**Table 1 cells-08-00401-t001:** Metabolic compounds and therapeutic approaches targeting breast cancer metabolism.

Metabolic targets	Agents or approaches	Effect on breast cancer cells	References
Arginine	ADI-PEG20	Autophagy-dependent cell death	[132]
Pyruvate metabolism	DCA	Re-establishment of mitochondrial function and OXPHOS metabolism^.^ mitochondrial dysfunction and apoptotic cell death	[134,135,136]
Pyruvate metabolism	DCA in combination with PDT	ICD	[139]
SLC6A14 transporter	α-MT	Inhibition of amino acid transporter. Apoptosis and autophagy	[157]
VDAC1	VDAC1 silencing	Cell growth arrest and tumour growth inhibition	[176]
Glutaminase	CB-839 Compound 968 BPTES	Inhibition of glutamine metabolism. Apoptotic cell death	[155,158,159]
FASN	FASN inhibition	Apoptotic cell death	[160]
OXPHOS	TAM Biguanides (Metformin, Phenformin)	Inhibition of Complex I of ETC and cytotoxicity. Increased glycolysis and pharmacological resistance induction.	[11,145,146,147,148,152,153,173,177]
OXPHOS	MitoTam	Inhibition of Complex I of ETC. Cancer cell death.	[179]
OXPHOS	ME344	Tumour growth inhibition	[173]
OXPHOS	FR58P1a	Metastatic capability reduction	[35]
OXPHOS and glycolysis	MGT	Anti-proliferative, anti-oxidant, and chemopreventive role	[140,141]
OXPHOS and glycolysis	TAM in combination with glycolysis inhibitor	Increased anti-tumour effect	[177]
OXPHOS and glycolysis	Metformin in combination with glycolysis inhibitor	Increased anti-tumour effect	[154]

DCA: dichloroacetate.

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
