# Peer review of "Mitochondrial Flexibility of Breast Cancers: A Growth Advantage and a Therapeutic Opportunity"

_cells, 2019, doi:10.3390/cells8050401_

Reviewer 1 Report

1)    Title: tumor strength; The authors should change this word to proper word.

2)    Abstract: delete “Therefore,”.

3)    P1, line 44: “getting in touch through”, please change to “due to”.

4)    Reference No. 1 seems include important information. However, Reference No. 1 is hard to obtain. The reviewer recommends make figure for the “1. Introduction, 1st paragraph”.

5)    “3. Breast Cancer Cells Mitochondrial Reprogramming” should be “Mitochondrial Reprogramming in Breast Cancer Cells”.

6)    Line 212: Reference [43]: Addition to [43], please cite the following article.

Indo, H. P.; Yen, H.-C.; Nakanishi, I.; Matsumoto, K.; Tamura, M.; Nagano, Y.; Matsui, H.; Gusev, O.; Cornette, R.; Okuda, T.; Minamiyama, Y.; Ichikawa, H.; Suenaga, S.; Oki, M.; Sato, T.; Ozawa, T.; St Clair, D. K.; Majima, H. J. A Mitochondrial Superoxide Theory for Oxidative Stress Diseases and Aging. J. Clin. Biochem. Nutr. 2015, 56, 1-7, doi: 10.3164/jcbn.14-42.

7)    Line 508, after [107]. Please add the following phrase, and cite a paper. “Indo et. demonstrated an evidence of signal transduction by mitochondrial ROS [* ].”

* Indo, H. P.; Hawkins, C. L.; Nakanishi, I.; Matsumoto, K.; Matsui, H.; Suenaga, S.; Davies, M. J.: St Clair, D. K.; Ozawa, T.; Majima, H. J. Role of Mitochondrial Reactive Oxygen Species in the Activation of Cellular Signals, Molecules and Function. In Pharmacology of Mitochondria; Handb Exp Pharmacol (HEP), Singh, H., Sheu S.-S., Eds.; Springer Nature Switzerland AG: Basel, Switzerland, 2017; Volume 240, pp. 439-456, doi: 10.1007/164_2016_117.

Author Response

Dear Reviewer,

thank you for the remarks and the comments concerning the review article entitled Mitochondrial Flexibility of Breast Cancers: a Tumour Strength and a Therapeutic Opportunity, (Manuscript ID: cells-469915) by Angelica Avagliano, Maria Rosaria Ruocco, Federica Aliotta, Immacolata Belviso, Antonello Accurso, Stefania Masone, Stefania Montagnani, Alessandro Arcucci.

Your comments were carefully considered and a point-by-point response is hereafter reported.

In particular, new added statements, statements relocated in the text, new legends figures and new added references are highlighted by using red colour, whereas deleted text is struck through and red highlighted. The numbers and the brackets of new added references are red highlighted in the text (for example [17]), whereas only the numbers of references relocated in the text are red highlighted (for example [42]). Furthermore, we have added three new figures (Figure 1), (Figure 4) and (Figure 5), whereas only the new numbers of the figures already present in the first version of the review article are red highlighted (Figure 2) and (Figure 3).

I hope that the revised version of the review article will be suitable for publication on Special Issue "Mitochondrial Metabolic Reprogramming and Nuclear Crosstalk in Cancer" of Cells.

POINT-BY-POINT RESPONSE

1) Title: tumor strength; The authors should change this word to proper word.

Response

We have changed the title of the review article as: Mitochondrial Flexibility of Breast Cancers: a Growth Advantage and a Therapeutic Opportunity.

2) Abstract: delete “Therefore,”.

Response

Yes, we have deleted “therefore” in the abstract.

3) P1, line 44: “getting in touch through”, please change to “due to”.

Response

We have not changed the sentence “getting in touch through” with “due to” at line 39 of new version, because “getting in touch” better indicates the crosstalk between cancer cells and tumour microenvironment.

4) Reference No. 1 seems include important information. However, Reference No. 1 is hard to obtain. The reviewer recommends make figure for the “1. Introduction, 1st paragraph”.

Response

We have made a new figure (Figure 1) illustrating the immunohistochemical properties and relative prognosis of the three mayor breast cancer types.

5) “3. Breast Cancer Cells Mitochondrial Reprogramming” should be “Mitochondrial Reprogramming in Breast Cancer Cells”.

Response

We have modified “ Breast Cancer Cells Mitochondrial Reprogramming” with “Mitochondrial Reprogramming in Breast Cancer Cells”.

6) Line 212: Reference [43]: Addition to [43], please cite the following article. Indo, H. P.; Yen, H.-C.; Nakanishi, I.; Matsumoto, K.; Tamura, M.; Nagano, Y.; Matsui, H.; Gusev, O.; Cornette, R.; Okuda, T.; Minamiyama, Y.; Ichikawa, H.; Suenaga, S.; Oki, M.; Sato, T.; Ozawa, T.; St Clair, D. K.; Majima, H. J. A Mitochondrial Superoxide Theory for Oxidative Stress Diseases and Aging. J. Clin. Biochem. Nutr. 2015, 56, 1-7, doi: 10.3164/jcbn.14-42.

Response

We have added the suggested article ([50] Indo, H. P.; Yen, H.-C.; Nakanishi, I.; Matsumoto, K.; Tamura, M.; Nagano, Y.; Matsui, H.; Gusev, O.; Cornette, R.; Okuda, T.; Minamiyama, Y.; Ichikawa, H.; Suenaga, S.; Oki, M.; Sato, T.; Ozawa, T.; St Clair, D. K.; Majima, H. J. A Mitochondrial Superoxide Theory for Oxidative Stress Diseases and Aging. J. Clin. Biochem. Nutr. 2015, 56, 1-7, doi: 10.3164/jcbn.14-42).

7) Line 508, after [107]. Please add the following phrase, and cite a paper. “Indo et. demonstrated an evidence of signal transduction by mitochondrial ROS [*].” * Indo, H. P.; Hawkins, C. L.; Nakanishi, I.; Matsumoto, K.; Matsui, H.; Suenaga, S.; Davies, M. J.: St Clair, D. K.; Ozawa, T.; Majima, H. J. Role of Mitochondrial Reactive Oxygen Species in the Activation of Cellular Signals, Molecules and Function. In Pharmacology of Mitochondria; Handb Exp Pharmacol (HEP), Singh, H., Sheu S.-S., Eds.; Springer Nature Switzerland AG: Basel, Switzerland, 2017; Volume 240, pp. 439-456, doi: 10.1007/164_2016_117.

Response

We have added at lines 672-673 the suggested sentence and the relative reference ([126] Indo, H. P.; Hawkins, C. L.; Nakanishi, I.; Matsumoto, K.; Matsui, H.; Suenaga, S.; Davies, M. J.: St Clair, D. K.; Ozawa, T.; Majima, H. J. Role of Mitochondrial Reactive Oxygen Species in the Activation of Cellular Signals, Molecules and Function. In Pharmacology of Mitochondria; Handb Exp Pharmacol (HEP), Singh, H., Sheu S.-S., Eds.; Springer Nature Switzerland AG: Basel, Switzerland, 2017; Volume 240, pp. 439-456, doi: 10.1007/164_2016_117).

In the complex, we have edited the English. In particular, we have corrected some grammar errors and we have made the text less convoluted.

Thank you for your kind attention.

Yours sincerely

Dr. Alessandro Arcucci

Department of Public Health

University of Naples Federico II

Naples, Italy

Via S. Pansini, 5

I-80131 Napoli (Italy)

Tel. +39-081-7463422

Fax: +39-081-7463409

Reviewer 2 Report

This is a very comprehensive review with several areas related to metabolic flexibility and mitochondrial functions covered. Figures 1 and 2 are very informative and appropriate to the review. 

There are some weaknesses that need to be addressed before publication:

Based on the title, “mitochondrial flexibility of breast cancers” is the main theme of the review.  Subsection 3 (Breast cancer cells mitochondrial reprogramming) (line 208 to line 396) is very detailed, but amorphous. Although Figure 2 depicts mitochondrial morphology alterations, there are no figures for the functional alterations of mitochondria. The authors should provide at least one figure to schematically show the major functional alterations of mitochondrial functions (along with the major connections to the rest of the cytoplasm and the nucleus.

The subsection 4, Mitochondrial Retrograde Regulation needs to be more comprehensive. For example, recent information on how retrograde metabolism impacts nuclear chromatin modifications (such as acetylation, methylation) needs to be included. What are mitochondrial pathways/molecules that feed into the nuclear epigenetic regulation? The first two sentences (lines 522-525) under subsection 5 appear to belong better under subsection 4. There should be at least one diagrammatic representation of the major mitochondrial retrograde events/pathways impinging on nuclear functions. 

Subsection 5 (Therapeutic Strategies Targeting Mitochondrial Alterations” is well written and comprehensive. 

There are several awkward sentences and grammatical errors scattered throughout the manuscript. I would suggest the authors seek the help of English editing. 

Especially of concern is the use of “strictly linked to…” phrases at multiple places. For example, sentence 206, “Moreover, p53 mutations induce alterations, strictly linked to immunological breast cancer types, in both glycolysis and OXPHOS”. It is not clear what the authors want to convey here. Whatever it is, it is incorrect to say that p53 mutations induce alterations that are strictly linked to immunological breast cancer types is incorrect. P53 mutations can cause various alterations in luminal, HER2-amplified, and triple negative immuno-histochemical subtypes of breast cancer, although p53 mutations are more prevalent in triple negative breast cancer.  Unless it is an absolute observation, it is better to avoid the usage “strictly…..”

Author Response

Dear Reviewer,

thank you for the remarks and the comments concerning the review article entitled Mitochondrial Flexibility of Breast Cancers: a Tumour Strength and a Therapeutic Opportunity, (Manuscript ID: cells-469915) by Angelica Avagliano, Maria Rosaria Ruocco, Federica Aliotta, Immacolata Belviso, Antonello Accurso, Stefania Masone, Stefania Montagnani, Alessandro Arcucci.

Your comments were carefully considered and a point-by-point response is hereafter reported.

In particular, new added statements, statements relocated in the text, new legends figures and new added references are highlighted by using red colour, whereas deleted text is struck through and red highlighted. The numbers and the brackets of new added references are red highlighted in the text (for example [17]), whereas only the numbers of references relocated in the text are red highlighted (for example [42]). Furthermore, we have added three new figures (Figure 1), (Figure 4) and (Figure 5), whereas only the new numbers of the figures already present in the first version of the review article are red highlighted (Figure 2) and (Figure 3).

I hope that the revised version of the review article will be suitable for publication on Special Issue "Mitochondrial Metabolic Reprogramming and Nuclear Crosstalk in Cancer" of Cells.

POINT-BY-POINT RESPONSE

1) There are some weaknesses that need to be addressed before publication:

Based on the title, “mitochondrial flexibility of breast cancers” is the main theme of the review.  Subsection 3 (Breast cancer cells mitochondrial reprogramming) (line 208 to line 396) is very detailed, but amorphous. Although Figure 2 depicts mitochondrial morphology alterations, there are no figures for the functional alterations of mitochondria. The authors should provide at least one figure to schematically show the major functional alterations of mitochondrial functions (along with the major connections to the rest of the cytoplasm and the nucleus.

Response

We have added a new figure (Figure 4) showing the functional alterations of mitochondria in breast cancer cells.

2) The subsection 4, Mitochondrial Retrograde Regulation needs to be more comprehensive. For example, recent information on how retrograde metabolism impacts nuclear chromatin modifications (such as acetylation, methylation) needs to be included. What are mitochondrial pathways/molecules that feed into the nuclear epigenetic regulation? The first two sentences (lines 522-525) under subsection 5 appear to belong better under subsection 4. There should be at least one diagrammatic representation of the major mitochondrial retrograde events/pathways impinging on nuclear functions.

Response

At lines 530-551 we have described in the new version of review article how retrograde metabolism impacts nuclear chromatin modifications (such as acetylation, methylation), and mitochondrial pathways/molecules feeding into the nuclear epigenetic regulation. Therefore, we have added the relative references ([102], [103], [104] and [105]). Moreover, we have relocated the first two sentences (lines 522-525) from subsection 5 to subsection 4 (lines 500-505) of the new version of the review article. We have added a new figure (Figure 5) showing the main cancerous pathways involved in Mitochondrial Retrograde Regulation.

3) There are several awkward sentences and grammatical errors scattered throughout the manuscript. I would suggest the authors seek the help of English editing.

Especially of concern is the use of “strictly linked to…” phrases at multiple places. For example, sentence 206, “Moreover, p53 mutations induce alterations, strictly linked to immunological breast cancer types, in both glycolysis and OXPHOS”. It is not clear what the authors want to convey here. Whatever it is, it is incorrect to say that p53 mutations induce alterations that are strictly linked to immunological breast cancer types is incorrect. P53 mutations can cause various alterations in luminal, HER2-amplified, and triple negative immuno-histochemical subtypes of breast cancer, although p53 mutations are more prevalent in triple negative breast cancer.  Unless it is an absolute observation, it is better to avoid the usage “strictly…..”

Response

We have tried to correct awkward sentences and grammatical errors through English editing. In particular, in the lines 261-262 we have deleted the statement “strictly linked to immunological breast cancer types” and furthermore we have removed the word “strictly” at multiple places of the text.

In the complex, we have edited the English. In particular, we have corrected some grammar errors and we have made the text less convoluted.

Thank you for your kind attention.

Yours sincerely

Dr. Alessandro Arcucci

Department of Public Health

University of Naples Federico II

Naples, Italy

Via S. Pansini, 5

I-80131 Napoli (Italy)

Tel. +39-081-7463422

Fax: +39-081-7463409

Reviewer 3 Report

In the review article ‘Mitochondrial Flexibility of Breast Cancers: a Tumour Strength and a Therapeutic Opportunity’ the authors take on themselves an uneasy task of assessing the role of mitochondria in breast cancer, a very broad subjects. While their piece contain some interesting pieces of information, it also neglects very important aspects as detailed below.

Major points:

1.      At several places throughout the manuscript the authors claim that OXPHOS is linked to ATP production (only). While this may be true in non-proliferative tissues, during cancer cell growth OXHPHOS has crucial biosynthetic functions. Its main role seems to be to synthesize aspartate (PMID:26232224., PMID:26232225) and to contribute to pyrimidine biosynthesis via the enzyme dihydroorotate dehydrogenase (PMID:30449682). In fact, breast cancer-derived tumors that completely lack ATP synthase are still able to produce tumors (PMID:30449682). This should definitely be corrected where applicable and relevant references should be included.

2.      It is strange that while the authors write about how mitochondrial function and OXPHOS are essential in breast cancer, they do not mention the seminal paper on this subject which demonstrates that breast cancer cells lacking mtDNA are able to form tumors only after new mtDNA has been acquired from tumor stroma and OXPHOS has been reconstituted (PMID: 25565207). There are also other paper that show, albeit not in breast cancer, that OXPHOS is necessary for tumorigenesis (PMID: 20421486)

3.      Why do authors discuss arginine auxothrophy, when from the point of view of mitochondria the aspartate/pyruvate auxothrophy is much more significant (lines 547-564)?

4.      When putting so much emphasis on tamoxifen as an inhibitor of CI, they should also discuss mitochondria-targeted tamoxifen, which is a much more efficient CI inhibitor than tamoxifen and a superior anticancer agent in this respect (PMID:27392540). BTW, the ability of tamoxifen to inhibit CI was first shown by Moreira et al (PMID:16410252), perhaps this should be also mentioned.

5.      The paper should be corrected for English, it is slightly convoluted in places.  

Minor points (in the order of appearance)

58.  “cancer cells represent a cell population principally non-tumorigenic [3].” This is perhaps an overstatement.

81. Please do not say “request of the energy”

88. Glycolysis definitely cannot synthesize all the cellular components required for growth.

196. p53 principally regulates Sco2 (PMID:16728594)

251. What is it “mitochondrial autophagy”?

364. Also this paper showed reductive carboxylation at the same time: PMID:22101433.

414 “metabolic cues or more direct routes” this sounds cool, but what are those direct routes? Could you specify?

528 “big energetic request” ??????

Table 1: Phenformin and metformin are analogous and both act at CI.

608  “It is known that the activity of mitochondrial Complex I is increased in breast cancer cells and can also modulate breast cancer progression [137].” Some place before is written that reduction of mtDNA makes breast cancer cells more tumorigenic. It seems that these statements are at odds. Maybe it  should not generalize this much?

626 “cell proliferation needs TCA energetic activity as primary font of biosynthetic precursors” ????????

633. “glutamine addiction is a common strategy used by cancer cells” Addiction is usually not a strategy, it is a liability.

642. CB-839 is not introduced at all. It suddenly pops up without anyone knowing from where.

666. Perhaps “directly” is meant instead of “straight”  “partecipates to regulation” ??????

681. “evidence” is uncountable

It seems to be going downhill with English at this point.

The text should be broken into smaller section, long as they are now these sections do not help readability

Author Response

Dear Reviewer,

thank you for the remarks and the comments concerning the review article entitled Mitochondrial Flexibility of Breast Cancers: a Tumour Strength and a Therapeutic Opportunity, (Manuscript ID: cells-469915) by Angelica Avagliano, Maria Rosaria Ruocco, Federica Aliotta, Immacolata Belviso, Antonello Accurso, Stefania Masone, Stefania Montagnani, Alessandro Arcucci.

Your comments were carefully considered and a point-by-point response is hereafter reported.

In particular, new added statements, statements relocated in the text, new legends figures and new added references are highlighted by using red colour, whereas deleted text is struck through and red highlighted. The numbers and the brackets of new added references are red highlighted in the text (for example [17]), whereas only the numbers of references relocated in the text are red highlighted (for example [42]). Furthermore, we have added three new figures (Figure 1), (Figure 4) and (Figure 5), whereas only the new numbers of the figures already present in the first version of the review article are red highlighted (Figure 2) and (Figure 3).

I hope that the revised version of the review article will be suitable for publication on Special Issue "Mitochondrial Metabolic Reprogramming and Nuclear Crosstalk in Cancer" of Cells.

POINT-BY-POINT RESPONSE

Major points:

1) At several places throughout the manuscript the authors claim that OXPHOS is linked to ATP production (only). While this may be true in non-proliferative tissues, during cancer cell growth OXHPHOS has crucial biosynthetic functions. Its main role seems to be to synthesize aspartate (PMID:26232224., PMID:26232225) and to contribute to pyrimidine biosynthesis via the enzyme dihydroorotate dehydrogenase (PMID:30449682). In fact, breast cancer-derived ‘'tumors that completely lack ATP synthase are still able to produce tumors (PMID:30449682). This should definitely be corrected where applicable and relevant references should be included.

Response

At lines 127-135 we have specified the role of OXPHOS in aspartate synthesis that sustains cells proliferation. In particular, we have summarized the different influences of pyrimidine biosynthesis, via enzyme dihydroorotate dehydrogenase, and ATP production, via ATP synthase, on tumorigenesis. Therefore, we have added the relative references ([20-22], [23]).

2) It is strange that while the authors write about how mitochondrial function and OXPHOS are essential in breast cancer, they do not mention the seminal paper on this subject which demonstrates that breast cancer cells lacking mtDNA are able to form tumors only after new mtDNA has been acquired from tumor stroma and OXPHOS has been reconstituted (PMID: 25565207). There are also other paper that show, albeit not in breast cancer, that OXPHOS is necessary for tumorigenesis (PMID: 20421486).

Response

At lines 124-127 we have mentioned the importance of mtDNA and mitochondrial respiration in tumorigenesis. Hence, we have added the relative reference ([19]).

3) Why do authors discuss arginine auxothrophy, when from the point of view of mitochondria the aspartate/pyruvate auxothrophy is much more significant (lines 547-564)?

Response

The sentences of lines 730-750 have highlighted the pivotal role of pyruvate-aspartate axis in tumor cell proliferation and survival during ETC dysfunction. Therefore, we have added the new relative references ([21], [133]).

4) When putting so much emphasis on tamoxifen as an inhibitor of CI, they should also discuss mitochondria-targeted tamoxifen, which is a much more efficient CI inhibitor than tamoxifen and a superior anticancer agent in this respect (PMID:27392540). BTW, the ability of tamoxifen to inhibit CI was first shown by Moreira et al (PMID:16410252), perhaps this should be also mentioned.

Response

At lines 982-997 we have described the higher anti-tumour activity of MitoTam compared to that of TAM. In particular, the cytotoxic effect of MitoTam is mitochondria-mediated because this compound selectively and strongly targets the CI. Therefore, we have added the relative reference ([179]).

At lines 966-968 we have mentioned the ability of tamoxifen in the inhibition of CI, including the reference of the paper of Moreira et al. ([178]).  

5) The paper should be corrected for English, it is slightly convoluted in places.  

Response

We have corrected the paper for English.

Minor points (in the order of appearance)

1) 58. “cancer cells represent a cell population principally non-tumorigenic [3].” This is perhaps an overstatement

Response

At lines 74-75 we have changed the statement “cancer cells represent a cell population principally non-tumorigenic [3]” with “cancer cells represent a cell population characterized by a low tumorigenic potential [3]”.

2) 81. Please do not say “request of the energy”

Response

At line 102 we have changed the statement “request of the energy” with “energy demand”.

3) 88. Glycolysis definitely cannot synthesize all the cellular components required for growth.

Response

In the line 111, the sentence “…synthesize all the cellular components required to grow” has been changed with “…synthesize many cellular components required to grow”.

4) 196. p53 principally regulates Sco2 (PMID:16728594).

Response

At line 247 we have added the reference [44] relative to PMID:16728594.

5) 251. What is it “mitochondrial autophagy”?

Response

At line 309 we have changed mitochondrial autophagy with mitophagy.

6) 364. Also this paper showed reductive carboxylation at the same time: PMID:22101433.

Response

At lines 454-457 we have shortly summarized the paper of Metallo et al. ([81]) describing the dependence of cancer cells on reductive glutamine carboxylation.

7) 414 “metabolic cues or more direct routes” this sounds cool, but what are those direct routes? Could you specify?

Response

At line 528 we have mentioned a direct route of MRR represented by mitochondria-related changes in intracellular Ca2+ homeostasis.

8) 528 “big energetic request” ??????

Response

At line 697 we have changed “big energetic request” with “increased energy demand”.

9) Table 1: Phenformin and metformin are analogous and both act at CI.

Response

We have brought together phenformin and metformin in Table 1.

10) 608 “It is known that the activity of mitochondrial Complex I is increased in breast cancer cells and can also modulate breast cancer progression [137].” Some place before is written that reduction of mtDNA makes breast cancer cells more tumorigenic. It seems that these statements are at odds. Maybe it  should not generalize this much?

Response

At lines 808-811 we have changed the statement “It is known that the activity of mitochondrial Complex I is increased in breast cancer cells and can also modulate breast cancer progression [137].” with “It is known that the aberrant activity of mitochondrial Complex I can modulate breast cancer progression affecting tumour growth and metastasis [151].”.

11) 626 “cell proliferation needs TCA energetic activity as primary font of biosynthetic precursors” ????????

Response

At lines 833-835 we have changed the statement “It is known that cell proliferation needs TCA energetic activity as primary font of biosynthetic precursors.” with “It is known that in proliferating cells TCA cycle is an important font of biosynthetic precursors [155].”

12) 633. “glutamine addiction is a common strategy used by cancer cells” Addiction is usually not a strategy, it is a liability.

Response

At line 843 we have changed “addiction” with “metabolism”.

13) 642. CB-839 is not introduced at all. It suddenly pops up without anyone knowing from where.

Response

At line 855 we have introduced the CB-839 compound.

14) 666. Perhaps “directly” is meant instead of “straight”  “partecipates to regulation” ??????

Response

At lines 884-885 we have changed “straight” with “directly” and “participates to regulation” with “contributes to regulation”.

15) 681. “evidence” is uncountable

Response

In the revision of the manuscript we have used “evidence” as an uncountable noun.

16) The text should be broken into smaller section, long as they are now these sections do not help readability

Response

It is too difficult to break the text into smaller sections since all the issues described in each subsection are interconnected.

In the complex, we have edited the English. In particular, we have corrected some grammar errors and we have made the text less convoluted. 

Thank you for your kind attention.

Yours sincerely

Dr. Alessandro Arcucci

Department of Public Health

University of Naples Federico II

Naples, Italy

Via S. Pansini, 5

I-80131 Napoli (Italy)

Tel. +39-081-7463422

Fax: +39-081-7463409

Round  2

Reviewer 3 Report

The paper has been substantially improved by the authors. There are some minor points left:

typo in Figure 1 caption ‘ Mayer Types’ should likely be ‘major types’

102 glycolysis does not supply ‘energy demand’. It can satisfy energy demand, or it can supply remaining energy.

134 ‘is not indispensable for tumorigenesis’: is dispensable for tumorigenesis’ ?

Fig. 4 – reductive carboxylation is NADPH dependent (performed by IDH1 and 2), NOT NADH-dependent (IDH3 is not involved) (see Mullen et al, ref. 80)

673 ‘demonstrated an evidence of’: Evidence is uncountable, should read ’present evidence for’ or ‘ Indo et al demonstrated …’, ‘showed’ etc. You cannot demonstrate evidence, but you can present it, find it, or you can have it

835 ‘important font of’ should be ‘important source of’ or something like that.

843 ‘utilization’ would make more sense than ‘metabolism’

Language editing is still advisable.

Author Response

Dear Reviewer,

thank you for the remarks and the comments concerning the review article entitled “Mitochondrial Flexibility of Breast Cancers: a Tumour Strength and a

Therapeutic Opportunity”, (Manuscript ID: cells-469915) by Angelica Avagliano, Maria Rosaria Ruocco, Federica Aliotta, Immacolata Belviso, Antonello Accurso, Stefania Masone, Stefania Montagnani, Alessandro Arcucci.

Your comments were carefully considered and a point-by-point response is hereafter reported.

In particular, changed statements and words are highlighted by using green colour, whereas deleted text is struck through and green highlighted. Furthermore, we have modified the Figure 4 and its caption.  

I hope that the revised version of the review article will be accepted for publication on Special Issue "Mitochondrial Metabolic Reprogramming and Nuclear Crosstalk in Cancer" of Cells.

POINT-BY-POINT RESPONSE

Reviewer 3

The paper has been substantially improved by the authors. There are some minor points left:

1) typo in Figure 1 caption ‘ Mayer Types’ should likely be ‘major types’

Response

In Figure 1 caption we have changed Mayor with Major.

2) 102 glycolysis does not supply ‘energy demand’. It can satisfy energy demand, or it can supply remaining energy.

Response

At line 102 we have changed “glycolysis supplies the remaining energy demand” with “glycolysis can satisfy energy demand “.

3) 134 ‘is not indispensable for tumorigenesis’: is dispensable for tumorigenesis’ ?

RESPONSE

At lines 134-135 we have changed “is not indispensable for tumorigenesis” with “is dispensable for tumorigenesis”.

4) Fig. 4 – reductive carboxylation is NADPH dependent (performed by IDH1 and 2), NOT NADH-dependent (IDH3 is not involved) (see Mullen et al, ref. 80)

RESPONSE

We have modified the Fig. 4 and its caption.  

5) 673 ‘demonstrated an evidence of’: Evidence is uncountable, should read ’present evidence for’ or ‘ Indo et al demonstrated …’, ‘showed’ etc. You cannot demonstrate evidence, but you can present it, find it, or you can have it.

Response

At line 676 we have changed “demonstrated an evidence of” with “presented evidence for”.

6) 835 ‘important font of’ should be ‘important source of’ or something like that.

Response

At line 838 we have changed “important font” with “important source”.

7) 843 ‘utilization’ would make more sense than ‘metabolism’.

Response

At line 846 we have modified “metabolism” with “utilization”

8) Language editing is still advisable.

Response

We have checked English

Thank you for your kind attention

Yours sincerely

Dr. Alessandro Arcucci

Department of Public Health

University of Naples Federico II

Naples, Italy

Via S. Pansini, 5

I-80131 Napoli (Italy)

Tel. +39-081-7463422

Fax: +39-081-7463409

Cells EISSN 2073-4409 Published by MDPI AG, Basel, Switzerland RSS E-Mail Table of Contents Alert
Back to Top